# Iron chelation improves ineffective erythropoiesis and iron overload in myelodysplastic syndrome mice

Wenbin An[1,2], Maria Feola[1], Maayan Levy[1], Srinivas Aluri[3], Marc Ruiz-Martinez[1], Ashwin Sridharan[3], Eitan Fibach[4], Xiaofan Zhu[2], Amit Verma[3], Yelena Ginzburg[1]*

[1]Division of Hematology and Medical Oncology, Tisch Cancer Institute, Icahn School of Medicine at Mount Sinai, New York, United States; [2]State Key Laboratory of Experimental Hematology, National Clinical Research Center for Blood Diseases, Division of Pediatric Blood Diseases Center, Institute of Hematology & Blood Diseases Hospital, Chinese Academy of Medical Sciences & Peking Union Medical College, Tianjin, China; [3]Division of Hematology and Medical Oncology, Albert Einstein College of Medicine, Bronx, United States; [4]Department of Hematology, Hadassah Medical Center, Hebrew University, Jerusalem, Israel

*For correspondence:
yelena.ginzburg@mssm.edu

**Abstract** Myelodysplastic syndrome (MDS) is a heterogeneous group of bone marrow stem cell disorders characterized by ineffective hematopoiesis and cytopenias, most commonly anemia. Red cell transfusion therapy for anemia in MDS results in iron overload, correlating with reduced overall survival. Whether the treatment of iron overload benefits MDS patients remains controversial. We evaluate underlying iron-related pathophysiology and the effect of iron chelation using deferiprone on erythropoiesis in NUP98-HOXD13 transgenic mice, a highly penetrant well-established MDS mouse model. Our results characterize an iron overload phenotype with aberrant erythropoiesis in these mice which was reversed by deferiprone-treatment. Serum erythropoietin levels decreased while erythroblast erythropoietin receptor expression increased in deferiprone-treated MDS mice. We demonstrate, for the first time, normalized expression of the iron chaperones *Pcbp1* and *Ncoa4* and increased ferritin stores in late-stage erythroblasts from deferiprone-treated MDS mice, evidence of aberrant iron trafficking in MDS erythroblasts. Importantly, erythroblast ferritin is increased in response to deferiprone, correlating with decreased erythroblast ROS. Finally, we confirmed increased expression of genes involved in iron uptake, sensing, and trafficking in stem and progenitor cells from MDS patients. Taken together, our findings provide evidence that erythroblast-specific iron metabolism is a novel potential therapeutic target to reverse ineffective erythropoiesis in MDS.

## Editor's evaluation

This study presents a valuable finding on the preclinical studies of chelating agents deferiprone (DFP) as potential therapeutics in Myelodysplastic Syndrome (MDS). The evidence supporting the claims of the authors is solid. The work will be of interest to clinicians or biologists working on MDS.

## Introduction

MDS is a heterogeneous group of bone marrow stem cell disorders characterized by ineffective hematopoiesis leading to blood cytopenias and increased incidence of transformation to acute myeloid leukemia (AML) (*Haferlach, 2019*; *Janssen et al., 1989*). In the United States, MDS affects approximately

100,000 people with a median age of 65 years at diagnosis and an incidence of 40/100,000 per year thereafter (*Rollison et al., 2008*; *Goldberg et al., 2010*). Most MDS patients suffer from the accumulating consequences of marrow failure compounded by other age-related diseases. Several categories of MDS patients are low-risk subtypes according to the revised International Prognostic Scoring System and have a longer median survival with the lowest rate of progression to AML (*Greenberg et al., 2012*; *Arber et al., 2016*). Low-risk MDS patients account for approximately two-thirds of all MDS patients with 30–50% requiring regular red blood cell (RBC) transfusions (*Dayyani et al., 2010*; *Kröger, 2019*). The main goals of therapy in low-risk MDS patients are to alleviate cytopenias and their associated symptoms and thereby improve quality of life (*Hellström-Lindberg, 2005*). RBC transfusions remain the mainstay of therapy in low-risk MDS (*Oliva et al., 2010*) and are the main source of progressive iron overload and consequent end-organ damage in transfusion-dependent patients. Because of the low rates of progression to AML in low-risk MDS patients, these patients have a substantial life expectancy and theoretically warrant screening for transfusional iron overload, known to increase morbidity and mortality in chronic RBC transfusion-dependent anemias (*Goldberg et al., 2010*; *Schafer et al., 1981*; *Greenberg et al., 2009*). Furthermore, although it is generally accepted that transfusional iron overload develops in low-risk transfusion-dependent MDS patients and methods to diagnose and treat iron overload are available, the risk-benefit ratio of treating iron overload in MDS patients remains controversial.

A correlation between iron overload and reduced survival has been demonstrated mostly by retrospective studies (*Malcovati et al., 2006*; *Malcovati et al., 2011*). RBC transfusion-dependence correlates strongly with decreased survival in MDS patients (*Garcia-Manero et al., 2008*; *Malcovati et al., 2005*), and MDS patients with elevated serum ferritin have significantly fewer Burst Forming Units (BFU-Es) but normal Granulocyte Macrophage Colony Forming Units (CFU-GM) (*Hartmann et al., 2013*). Iron overload has been shown to inhibit BFU-E colony formation and erythroblast differentiation in both murine and human hematopoietic progenitors in vitro (*Taoka et al., 2012*). Finally, cells exposed to excess iron exhibit dysplastic changes with increased intracellular reactive oxygen species (ROS) and decreased expression of anti-apoptotic genes (*Taoka et al., 2012*), and the addition of iron to MDS patients' peripheral blood mononuclear cells resulted in increased ROS and DNA damage, triggering apoptosis (*Fibach and Rachmilewitz, 2012*; *Pan et al., 1999*). These data suggest that iron overload may lead to a deleterious effect on hematopoiesis, worsening disease in MDS (*Camaschella et al., 2007*).

Along these lines, more recent studies demonstrate the potential benefit of iron chelation therapy on the overall survival in low-risk MDS patients. Iron chelation is associated with improved hemoglobin (Hb) and reduced RBC transfusion requirements in some patients (*Oliva et al., 2010*; *Badawi et al., 2010*). Most recently, the TELESTO trial demonstrated prolonged event-free survival in iron overloaded lower risk MDS patients treated with iron chelation (i.e. deferasirox) (*Angelucci et al., 2020*). However, no clear improvement in Hb or reduction in RBC transfusions and no effect on overall survival was observed in deferasirox-treated MDS patients. We anticipate that the lack of effect of deferasirox on erythropoiesis is a consequence of the specific iron chelator selected. Alternatively, unlike other commercially available iron chelators, deferiprone (DFP) has a lower iron binding affinity relative to transferrin (*Sohn et al., 2008*). As a consequence, the DFP iron chelating effect enables iron transfer from parenchymal cells to increase transferrin saturation which results in increase hepcidin expression in the liver in response to systemic iron (*Schmidt et al., 2015*; *Casu et al., 2016*). The physiologic effect of increased hepcidin responsiveness in turn prevents further iron absorption and recycling, decreasing iron availability for erythropoiesis. On the foundation of this knowledge, we hypothesize that in addition to decreasing the progression of systemic iron overload, DFP could also ameliorate ineffective erythropoiesis in MDS. A direct beneficial effect of DFP on erythropoiesis in MDS has yet to be demonstrated.

Here, we evaluate the effect of iron chelation on erythropoiesis in NUP98-HOXD13 transgenic (NHD13) mice as a well-established in vivo model of MDS. We use NHD13 mice because they replicate alterations in gene expression identified in MDS patients and are the most commonly used transgenic mouse models of MDS (*Liu et al., 2022*). More specifically, NHD13 mice are characterized by abortive erythroblast maturation and consequent ineffective erythropoiesis and are, therefore, of target interest in our search for a mechanistic understanding of ineffective erythropoiesis to target its amelioration (*Lin et al., 2005*; *Slape et al., 2008*). Lastly, NHD13 mice have been successfully used in

**Table 1.** Hematopoiesis- and Iron-related Characteristics of NHD13 mice.

| | RBC count | Hb | MCV | Retic count | WBC count | Platelet count | Serum EPO | Bone marrow cells | Liver iron concentration |
|---|---|---|---|---|---|---|---|---|---|
| (units) | $(10^6 /\mu L)$ | (g/dL) | (fL) | $(10^6 /\mu L)$ | $(10^6 /\mu L)$ | $(10^3 /\mu L)$ | $(\mu g/\mu L)$ | $(10^7$ cells) | (mg/g dry weight) |
| WT | 10±0.1 | 14.7±0.1 | 50±0.5 | 495±67 | 5.5±0.6 | 759±67 | 308±48 | 12.4±0.7 | 0.23±0.07 |
| MDS | 6.73±0.29 | 11.2±0.4 | 60±1.3 | 440±30 | 3.0±0.4 | 672±80 | 4832±1,154 | 15±0.8 | 0.51±0.04 |
| | *** | *** | *** | NS | ** | NS | *** | * | * |

WT = wild type; MDS = myelodysplastic syndrome (NHD13) mice; RBC = red blood cell; Hb = hemoglobin; MCV = mean corpuscular hemoglobin; Retic = reticulocyte; WBC = white blood cell; EPO = erythropoietin; NS = not significant; * $P<0.05$; ** $P<0.01$; *** $P<0.0001$.

pre-clinical studies to investigate the effects of an activin receptor II ligand trap on reversing ineffective erythropoiesis in MDS mice (*Suragani et al., 2014*).

Our results demonstrate that NHD13 mice exhibits anemia, increased serum erythropoietin (EPO), expanded erythropoiesis in the bone marrow and spleen, and parenchymal iron overload, consistent with a low-risk MDS phenotype in patients. In addition, we demonstrate decreased EPO-responsiveness, decreased erythroblast differentiation, and impaired enucleation in bone marrow erythroblasts from MDS mice. Furthermore, iron chelation with DFP, in addition to decreasing parenchymal iron deposition, restores hepdicin:iron responsiveness, partially reverses anemia, and normalizes serum EPO concentration in MDS mice. DFP-treated MDS mice also exhibit normalized erythroblast differentiation, expression of *Gata1* and *Epor* (EPO receptor) as well as that of iron chaperones *Pcbp1* and *Ncoa4*, and increased erythroblast ferritin concentration in bone marrow erythroblasts. Finally, we demonstrate aberrant expression of genes involved in iron uptake, sensing, and trafficking in MDS patient bone marrow stem and progenitor cells. Taken together, our data for the first time provides in vivo evidence that ineffective erythropoiesis in MDS mice is responsive to iron chelation with DFP, normalizing erythroblast iron trafficking and restoring EPO responsiveness to reverse anemia in MDS.

## Results
### MDS mice exhibit elevated mcv anemia with expanded erythropoiesis and iron overload, appropriate as a model of low-risk MDS

To establish the expected MDS phenotype, mice were sacrificed at 6 months of age and demonstrate significantly reduced RBC counts and Hb concentration, increased MCV, and no difference in reticulocyte count relative to WT controls (*Table 1*). MDS mice also exhibit decreased white blood cell (WBC) count and no difference in platelet count (*Table 1*). Consistently, our experiments also demonstrate that MDS mice exhibit significantly increased serum EPO concentration and increased bone marrow cellularity (*Table 1*). Furthermore, we demonstrate borderline increased apoptosis in bone marrow erythroblast in MDS mice, reaching significance in OrthoE (*Figure 1—figure supplement 1*). Finally, liver iron concentration is significantly increased in MDS mice (*Table 1*). In conjunction with previously published work (*Lin et al., 2005*; *Slape et al., 2008*; *Suragani et al., 2014*), this combination of characteristics provides substantial evidence that NHD13 mice are an appropriate mouse model for MDS (*Cui et al., 2014*).

### Iron chelation with DFP reverses iron overload, normalized erythroferrone expression in bone marrow erythroblasts, and improves hepcidin iron responsiveness in MDS mice

Next, we evaluate the effects of iron chelation with DFP in 5-month-old MDS mice, treated for 4 weeks with DFP and sacrificed for analyses at 6 months of age. First, we demonstrate that DFP can be detected in the serum of DFP-treated mice (*Figure 1—figure supplement 2*). Second, our results demonstrate that DFP-treated MDS mice exhibit increased serum iron concentration and transferrin saturation (*Figure 1A and B*); both male and female mice demonstrate equivalent responses to DFP (*Figure 1—figure supplement 3*). These findings are consistent with the reversal of increased parenchymal iron loading in MDS mice after DFP treatment, exhibiting decreased liver, spleen, and bone marrow non-heme iron concentration (*Figure 1C–E*) and validate the expected re-distribution of iron

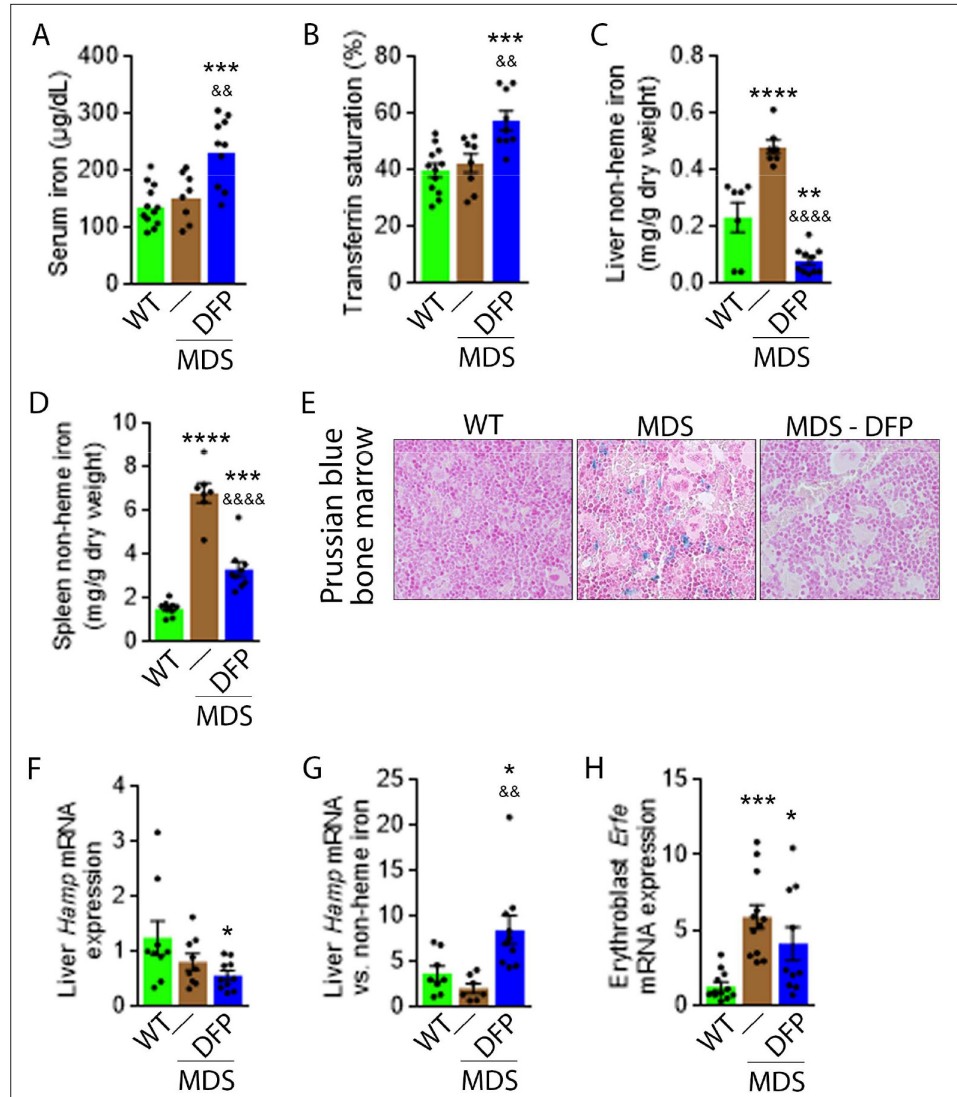

**Figure 1.** DFP reverses parenchymal iron overload and restores hepcidin iron responsiveness in MDS mice. DFP results in increased serum iron (**A**) and transferrin saturation (**B**) while reducing parenchymal iron in the liver, spleen, and bone marrow (**C-E**). While liver *Hamp* mRNA expression is unchanged in WT, MDS, and DFP-treated MDS mice (**F**), *Hamp* responsiveness to iron is normalized in DFP-treated MDS mice (**G**) (n=7–10 mice/group). (**H**) DFP results in more normal *Erfe* mRNA expression (n=10–12 mice/group) in sorted bone marrow erythroblasts from MDS mice analyzed after 1 month of treatment. *p<0.05 vs. WT; **p<0.01 vs. WT; ***p<0.001 vs. WT; ****p<0.0001 vs. WT; &p<0.05 vs. MDS; &&p<0.01 vs. MDS; &&&&p<0.0001 vs. MDS; Abbreviations: WT = wild type; MDS = myelodysplastic syndrome; DFP = deferiprone; Hamp = hepcidin; Erfe = erythroferrone.

The online version of this article includes the following source data and figure supplement(s) for figure 1:

**Source data 1.** Source data for iron-related parameters in wild type (WT), myelodysplastic syndrome (MDS), and DFP-treated MDS mice.

**Figure supplement 1.** Erythroblast apoptosis in MDS mice.

**Figure supplement 1—source data 1.** Source data for flow analysis of apoptosis measured by activated caspase 3/7 in bone marrow erythroblasts from wild type (WT) and myelodysplastic syndrome (MDS) mice.

**Figure supplement 2.** Quantification of serum DFP concentration in DFP-treated WT and MDS mice.

**Figure supplement 2—source data 1.** Source data for serum deferiprone (DFP) concentration in DFP-treated wild type (WT) and myelodysplastic syndrome (MDS) mice.

**Figure supplement 3.** DFP resulted in similar effects on transferrin saturation in male and female MDS mice.

**Figure supplement 3—source data 1.** Source data for transferrin saturation in male and female myelodysplastic

*Figure 1 continued*

syndrome (MDS) and DFP-treated MDS mice.

**Figure supplement 4.** Bone marrow erythroblast ferritin is increased in DFP-treated MDS mice.

**Figure supplement 4—source data 1.** Western blots with ferritin H antibody staining relative to actin in bone marrow erythroblast enriched CD45 negative cells from wild type (WT), myelodysplastic syndrome (MDS), and DFP-treated MDS mice.

**Figure supplement 4—source data 2.** Source data for quantification of ferritin heavy chain (FTH1) protein concentration relative to actin in bone marrow erythroblast enriched CD45 negative cells from wild type (WT), myelodysplastic syndrome (MDS), and DFP-treated MDS mice.

**Figure supplement 5.** Effects of DFP on the liver STAT3 expression in MDS mice.

**Figure supplement 5—source data 1.** Western blots with STAT3 and pSTAT3 antibody staining relative to GAPDH in liver from wild type (WT), myelodysplastic syndrome (MDS), and DFP-treated MDS mice.

**Figure supplement 5—source data 2.** Source data for *Saa1* in liver from wild type (WT), myelodysplastic syndrome (MDS), and DFP-treated MDS mice.

**Figure supplement 6.** Effect of DFP on erythroblast *Erfe* expression in WT mice.

**Figure supplement 6—source data 1.** Source data for erythroferrone (*Erfe*) in sorted bone marrow erythroblasts from wild type (WT) and DFP-treated WT mice.

from parenchymal deposition to the circulating compartment, to ultimately enable excretion. Furthermore, ferritin concentration is not statistically significantly different in MDS or DFP-treated MDS bone marrow erythroblast-rich CD45 negative cells (*Figure 1—figure supplement 4*). Finally, *Hamp* expression, the gene encoding for hepcidin, in the liver is unchanged and *Hamp*-iron responsiveness is decreased in MDS relative to WT mice (*Figure 1F and G*). While *Hamp* expression is not increased in the liver of DFP-treated MDS mice (*Figure 1F*), it is significantly increased relative to non-heme iron concentration in the liver (*Figure 1G*) providing evidence of enhanced hepcidin responsiveness to iron in DFP-treated MDS mice.

Hepcidin expression represents the effect of multiple pathways. Specifically, hepcidin is upregulated in response to liver iron stores and circulating iron; increased in the setting of inflammation; and downregulated in conditions of expanded or ineffective erythropoiesis as a consequence of elevated *Erfe* expression, the gene name for erythroferrone (ERFE), in erythroblasts (*Ginzburg, 2019*). Our results demonstrate that DFP-treated MDS mice exhibit decreased liver iron concentration (*Figure 1C*) while increasing transferrin saturation (*Figure 1B*) and no evidence of inflammation-mediated signaling to hepcidin in DFP-treated MDS mouse liver (*Figure 1—figure supplement 5*). Thus, the effect of iron redistribution on hepcidin expression in DFP-treated MDS mice is predictably small, leading us to also evaluate the contribution of changes in erythropoiesis on increased hepcidin responsiveness in DFP-treated MDS mice.

Erythroblast *Erfe* expression is increased in MDS mouse bone marrow (*Figure 1H*), the response expected in the setting of increased serum EPO concentration (*Kautz et al., 2014*) in MDS relative to WT mice. Similar to other diseases of ineffective erythropoiesis, increased expression of bone marrow *Erfe* is expected to suppress hepcidin and decrease hepcidin iron responsiveness, resulting in iron overload (*Kautz et al., 2014*; *Kautz et al., 2015*). As a consequence, our current findings demonstrate that increased bone marrow erythroblast *Erfe* expression results in inappropriately low liver *Hamp* expression relative to parenchymal iron loading (*Figure 1C–G*), demonstrating decreased *Hamp* responsiveness to iron in MDS mice. These findings further support the use of these mice as an appropriate MDS model in which decreased hepcidin iron responsiveness is a consequence of expanded erythropoiesis, leading to systemic iron overload observed in this disease (*Cui et al., 2014*). Furthermore, the reversal of ineffective erythropoiesis in DFP-treated MDS mice, with partial normalization of *Erfe* expression (*Figure 1H*), correlates with restored *Hamp* responsiveness to iron and reversal of parenchymal iron loading in the liver, spleen, and bone marrow (*Figure 1C–G*). Finally, erythroblast *Erfe* expression is borderline decreased also in DFP-treated WT mice (*Figure 1—figure supplement 6*). Taken together, our results provide further evidence that mitigating the hepcidin pathway is relevant in the pathophysiology of ineffective erythropoiesis and its reversal.

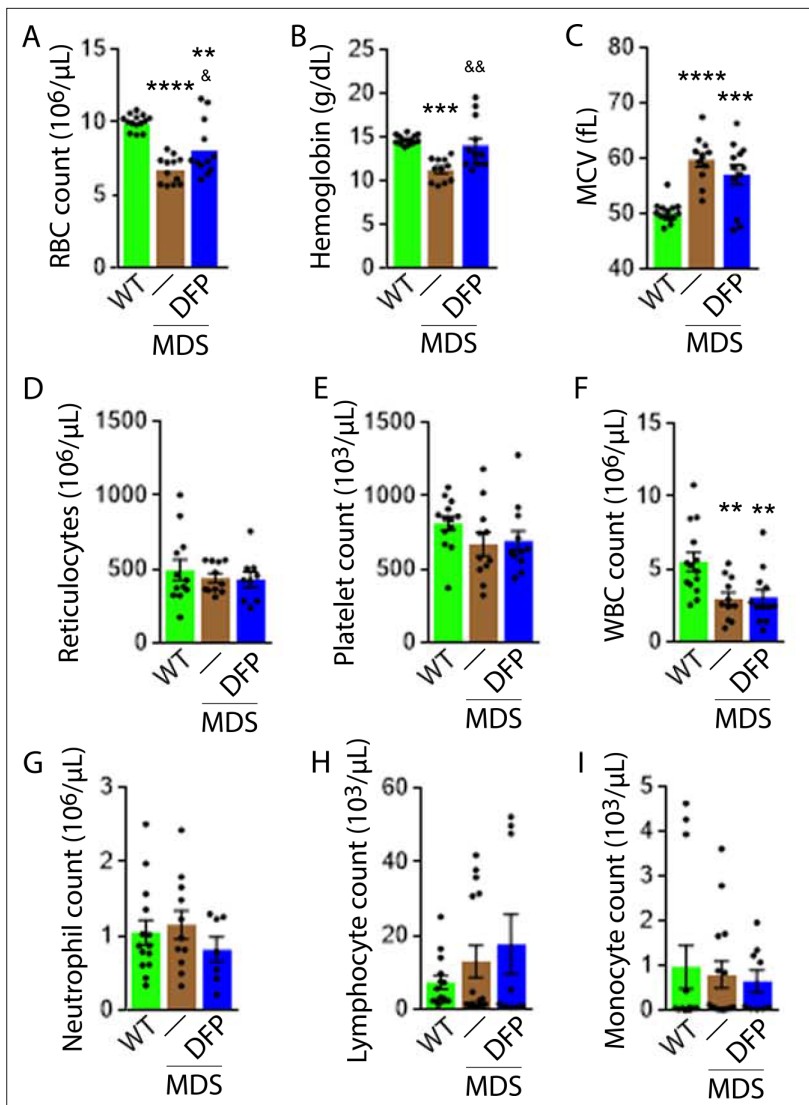

**Figure 2.** Elevated MCV anemia in MDS mice is partially reversed by DFP. Circulating RBC count (**A**), hemoglobin (**B**), MCV (**C**), reticulocyte count (**D**), platelet count (**E**), WBC count (**F**), neutrophil count (**G**), lymphocyte count (**H**), and monocyte count (**i**) in WT, MDS, and DFP-treated MDS mice (n=10–14 mice/group) analyzed after 1 month of treatment. *$p < 0.05$ vs. WT; **$p < 0.01$ vs. WT; ***$p < 0.001$ vs. WT; ****$p < 0.0001$ vs. WT; &$p < 0.05$ vs. MDS; &&$p < 0.01$ vs. MDS. Abbreviations: WT = wild type; MDS = myelodysplastic syndrome; DFP = deferiprone; RBC = red blood cell; MCV = mean corpuscular volume; WBC = white blood cell.

The online version of this article includes the following source data and figure supplement(s) for figure 2:

**Source data 1.** Source data for circulating cell counts and other parameters in wild type (WT), myelodysplastic syndrome (MDS), and DFP-treated MDS mice.

**Figure supplement 1.** DFP resulted in similar effects on hemoglobin in male and female myelodysplastic syndrome (MDS) mice.

**Figure supplement 1—source data 1.** Source data for hemoglobin in male and female myelodysplastic syndrome (MDS) and DFP-treated MDS mice.

## Iron chelation with DFP improves ineffective erythropoiesis in MDS mice

DFP-treated MDS mice exhibit increased Hb and RBC count relative to untreated MDS mice and no change in MCV or reticulocytosis (*Figure 2A–D*); both male and female mice demonstrate approximately equivalent responses to DFP (*Figure 2—figure supplement 1*). In addition, the WBC count

remained low while platelet, neutrophil, lymphocyte, and monocyte counts were unchanged in DFP-treated relative to untreated MDS mice (*Figure 2E–I*). Furthermore, while the spleen size increased in MDS relative to WT mice and is not significantly decreased after DFP (*Figure 3A*), splenic architecture is improved (*Figure 3B*)—with relatively decreased red pulp and more organized splenic nodules—and serum EPO concentration is normalized (*Figure 3C*) in DFP-treated relative to untreated MDS mice. Consistently, the total number of erythroblasts and the erythroid fraction in the bone marrow are increased in MDS relative to WT mice and normalized in DFP-treated relative to untreated MDS mice, with evident decrease, especially in BasoE and PolyE fractions (*Figure 3D–F*); both male and female mice demonstrate similar responses to DFP (*Figure 3—figure supplement 1*). DFP also leads to a normalized bone marrow erythroblast differentiation in MDS mice (*Figure 3G*) with a proportionally increased PolyE and decreased OrthoE fractions in bone marrow from MDS mice, normalized after DFP treatment, consistent with a block of erythroblast differentiation at PolyE in MDS patients (*Ali et al., 2018*). Finally, erythroblast apoptosis is unchanged (*Figure 3H*) despite decreased erythroblast ROS in DFP-treated relative to untreated MDS mice (*Figure 3I*). These findings are globally consistent with the improvement in ineffective erythropoiesis in response to DFP treatment without effects on erythroblast apoptosis in MDS.

Importantly, we evaluate the effects of DFP in WT mice. Specifically, our results demonstrate no change in RBC count, Hb, MCV, reticulocyte count, or serum EPO in DFP-treated relative to untreated WT mice (*Figure 3—figure supplement 2*). In addition, the bone marrow erythroblast fraction in DFP-treated WT mice is decreased, especially in the late stages of terminal erythropoiesis (i.e. PolyE and OrthoE), similar to DFP-treated MDS mice. However, unlike DFP-treated MDS mice, erythroid differentiation is decreased in DFP-treated relative to untreated WT mice (*Figure 3—figure supplement 3*). Furthermore, similar to DFP-treated MDS mice, DFP in WT mice results in unchanged erythroblast apoptosis albeit without affecting ROS (*Figure 3—figure supplement 4*). Taken together, these findings support our conclusions that reversal of ineffective erythropoiesis in MDS mice occurs independently of changes in erythroblast apoptosis, that DFP has a direct effect on erythropoiesis, and that the differences in effect on MDS and WT mice are dependent on the effectiveness of erythropoiesis in the underlying state. Finally, we confirmed that the effect of DFP in WT mice occurs despite a relatively higher serum DFP concentration and metabolized DFP-G concentration in WT relative to MDS mice (*Figure 3—figure supplement 5*).

## Normalized expression of EPO downstream genes in bone marrow erythroblast from DFP-treated MDS mice

To evaluate erythropoiesis more closely, we also measure *Gata1* and *Bcl11a* expression to assess the effect of DFP on other EPO-STAT5 target genes in MDS erythroblasts. *Gata1* and *Bcl11a* expression is known to be downstream of EPO. While GATA1 is an important transcriptional regulator in normal erythropoiesis (*Ling and Crispino, 2020*), BCL-xl is implicated in the anti-apoptosis effect of EPO on erythroblasts (*Koulnis et al., 2012*). Aberrant GATA1 expression in MDS has been described with evidence of increased GATA1 expression in bone marrow CD34 + stem and progenitor cells as well as CD71 + erythroblasts from MDS patients (*Maratheftis et al., 2007*). Furthermore, normal upregulation of GATA1 and BCL-xl during human erythroid differentiation is lost in MDS (*Hopfer et al., 2012*). Neither *Gata1* nor *Bcl11a* expression has previously been evaluated in MDS mice; their expression is expected to increase in conditions of elevated EPO concentration.

Our results demonstrate that *Gata1* mRNA expression is increased in sorted bone marrow ProE, borderline decreased in BasoE, and decreased in PolyE and OrthoE erythroblasts from MDS relative to WT mice (*Figure 4A–D*), consistent with expectations that GATA1 expression is elevated in MDS patient bone marrow stem and progenitor and early erythroblasts (*Maratheftis et al., 2007*) with loss of upregulation during erythroblast differentiation (*Hopfer et al., 2012*). DFP treatment restores *Gata1* mRNA expression relative to untreated MDS or WT mice (*Figure 4A–D*). Furthermore, *Bcl11a* mRNA expression is decreased in bone marrow erythroblasts from MDS relative to WT mice and returns to normal expression levels in DFP-treated MDS mice (*Figure 4F and G*) despite increased serum EPO (*Figure 3C*) and borderline increased erythroblast apoptosis (*Figure 3H*) in MDS erythroblasts, normalized in DFP-treated relative to untreated MDS mice. Bone marrow erythroblast *Bcl11a* expression is also increased in DFP-treated WT mice (*Figure 4—figure supplement 1*). These results raise an important question, namely whether physiological or pathophysiological nuances in EPO-STAT5

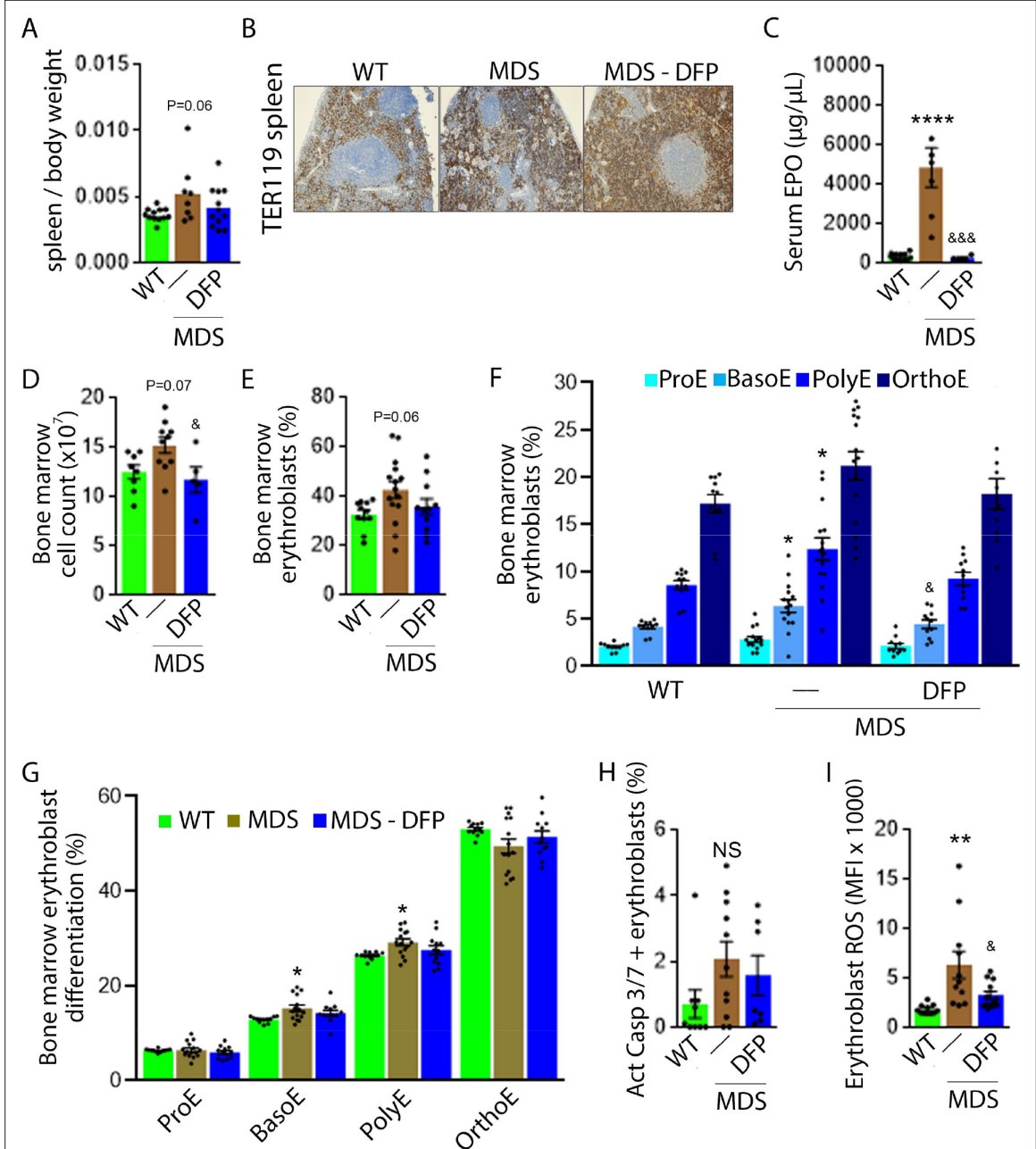

**Figure 3.** Expanded and ineffective erythropoiesis in MDS mice is partially reversed by DFP. Spleen weight (n=11–12 mice/group) (**A**), splenic architecture (n=5 mice/group) (**B**), serum EPO concentration (n=5–12 mice/group) (**C**), bone marrow erythroblast count (n=13–15 mice/group) (**D**), and the total fraction of erythroblasts in the bone marrow (n=13–15 mice/group) (**E**) are more normal in DFP-treated MDS mice analyzed after 1 month of treatment. The fraction of all stages of terminal erythropoiesis is increased in MDS relative to WT mice in BasoE and PolyE stages and decreased in DFP-treated relative to untreated MDS mice in BasoE stages (n=13–15 mice/group) (**F**). Erythroblast differentiation in the bone marrow, decreased in MDS relative to WT, is normalized in DFP-treated relative to untreated MDS mice (n=13–15 mice/group) (**G**). In addition, erythroblast apoptosis, as measured by activated caspase 3/7, is unchanged in DFP-treated MDS mice (n=7–11 mice/group) (**H**). Finally, erythroblast ROS is decreased in DFP-treated relative to untreated MDS mice (n=11–12 mice/group) (**i**) analyzed after 1 month of treatment. *p<0.05 vs. WT; **p<0.01 vs. WT; ****p<0.0001 vs. WT; &p<0.05 vs. MDS; &&&p<0.001 vs. MDS; Abbreviations: WT = wild type; MDS = myelodysplastic syndrome; DFP = deferiprone; EPO = erythropoietin; Act casp 3/7 = activated caspase 3 and 7; ROS = reactive oxygen species; ProE = pro-erythroblasts; BasoE = basophilic erythroblasts; PolyE = polychromatophilic erythroblasts; OrthoE = orthochromatophilic erythroblasts; NS = not significant.

The online version of this article includes the following source data and figure supplement(s) for figure 3:

**Source data 1.** Source data for erythropoiesis-related parameters in serum, bone marrow, and spleen from wild type (WT), myelodysplastic syndrome (MDS), and DFP-treated MDS mice.

*Figure 3 continued on next page*

*Figure 3 continued*

**Figure supplement 1.** DFP resulted in similar effects on erythropoiesis in male and female myelodysplastic syndrome (MDS) mice.

**Figure supplement 1—source data 1.** Source data for bone marrow erythroblasts in male and female myelodysplastic syndrome (MDS) and DFP-treated MDS mice.

**Figure supplement 2.** DFP has no effect on circulating red blood cell parameters and serum erythropoietin in WT mice.

**Figure supplement 2—source data 1.** Source data for circulating red blood cell parameters and serum erythropoietin from wild type (WT) and DFP-treated WT mice.

**Figure supplement 3.** Effect of DFP on erythropoiesis in WT mice.

**Figure supplement 3—source data 1.** Source data for total bone marrow erythroblasts from wild type (WT) and DFP-treated WT mice.

**Figure supplement 3—source data 2.** Source data for bone marrow ProE, BasoE, PolyE, and OrthoE erythroblasts from wild type (WT) and DFP-treated WT mice.

**Figure supplement 4.** DFP has no effect on erythroblast apoptosis and reactive oxygen species in WT mice.

**Figure supplement 4—source data 1.** Source data for bone marrow erythroblast apoptosis as measured by activated caspase 3/7 and ROS from wild type (WT) and DFP-treated WT mice.

**Figure supplement 5.** Quantification of serum DFP-glucuronide concentration in DFP-treated WT and MDS mice.

**Figure supplement 5—source data 1.** Source data for serum DFP and G-DFP concentrations from DFP-treated wild type (WT) and MDS mice.

**Figure supplement 6.** Gating strategy for delineating erythroblasts in mouse bone marrow.

---

signaling can conceptually separate EPO responsiveness from EPO-mediated anti-apoptotic effects in erythroblasts, similar to a hypothesis alluded to in prior publication (*Fontenay-Roupie et al., 1999*).

We then evaluate signaling pathways downstream of EPO. Both STAT5 and AKT signaling are essential for erythropoiesis. Prior work demonstrates that the expected STAT5 signaling response to EPO is hampered by iron restriction (*Khalil et al., 2018*). Others demonstrate that AKT signaling is implicated in EPO-mediated erythroblast survival (*Ghaffari et al., 2006*), essential in conditions with elevated EPO when *Epor* expression is suppressed (*Bullock et al., 2010*; *Nai et al., 2015*; *Zhao et al., 2016a*). We demonstrate STAT5 and AKT phosphorylation in bone marrow erythroblasts does not change in DFP-treated relative to untreated MDS mice (*Figure 4—figure supplement 2*). These findings suggest that the expected changes in signaling downstream of EPO are unaffected by DFP administration, implicating possible changes in erythroblast *Epor* expression in DFP-treated MDS mice.

## DFP increases *Epor* expression in later stage MDS erythroblasts

Next, we explore erythroblast *Epor* expression in DFP-treated and untreated MDS mice. We hypothesize that *Epor* plays an important role in EPO responsiveness that is independent of EPO concentration. This hypothesis is based on findings in EpoR-H mice, a knock-in mutation leading to normal EPO-EpoR binding and signaling but absent EpoR internalization and degradation (*Menon et al., 2006*; *Becker et al., 2010*; *Sulahian et al., 2009*). These mice exhibit decreased serum EPO levels, elevated RBC counts, and a smaller proportion of mature erythroid precursors in the bone marrow relative to WT mice (*Khalil et al., 2018*), suggesting that EpoR expression may influence erythroblast differentiation in a manner that is complementary to the anti-apoptotic effect of EPO.

Based on this premise, we anticipate that *Epor* expression is decreased in bone marrow erythroblast from MDS relative to WT mice, and restored in DFP-treated relative to untreated MDS mice. We used sorted bone marrow to evaluate *Epor* expression in progressive stages of terminal erythropoiesis. First, *Epor* expression is borderline increased in ProE and significantly increased in BasoE, earlier stage erythroblasts (*Figure 5A and B*) and decreased in PolyE and OrthoE, later stage erythroblasts (*Figure 5C and D*), in MDS relative to WT mice. Second, *Epor* expression is significantly increased in all-stage erythroblasts in DFP-treated relative to untreated MDS or WT mice (*Figure 5A–D*). Finally, DFP does not result in increased *Epor* expression in bone marrow erythroblasts from WT mice (*Figure 5—figure supplement 1*). Taken together, increased later-stage erythroblast *Epor* expression is a potential mechanism by which DFP leads to enhanced EPO responsiveness and enhanced erythroid differentiation despite decreased serum EPO concentration, reversing ineffective erythropoiesis exclusively in MDS mice.

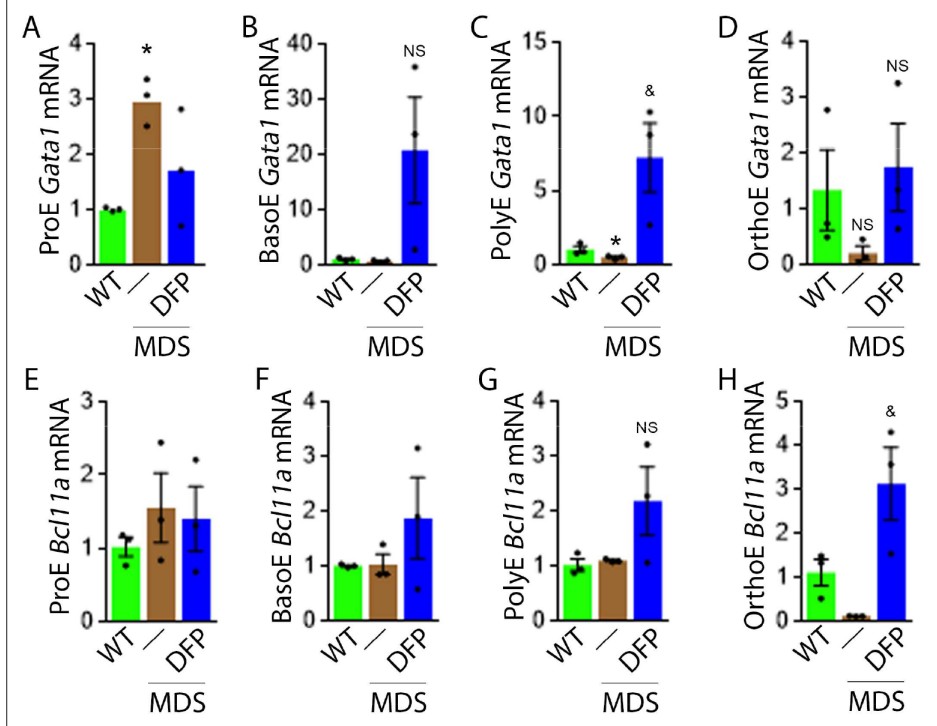

**Figure 4.** DFP leads to normalized gene expression downstream of EPO in MDS erythroblasts. *Gata1* mRNA expression is increased in sorted bone marrow ProE (**A**), unchanged in BasoE (**B**), and significantly decreased in PolyE (**C**) erythroblasts from MDS relative to WT mice; DFP treatment restores *Gata1* mRNA expression relative to untreated MDS or WT mice in all including OrthoE (**D**) erythroblasts (n=15–21 mice/group). DFP treatment in MDS mice does not affect *Bcl11a* mRNA expression in sorted bone marrow ProE (**E**), BasoE (**F**), and PolyE (**G**), and increases it in OrthoE (**H**) erythroblasts relative to untreated MDS or WT mice (n=10–12 mice/group). *p<0.05 vs. WT; &p<0.05 vs. MDS; Abbreviations: WT = wild type; MDS = myelodysplastic syndrome; DFP = deferiprone; Gata1 = erythroid transcription factor; Bcl11a B cell lymphoma 11 a; ProE = pro-erythroblasts; BasoE = basophilic erythroblasts; PolyE = polychromatophilic erythroblasts; OrthoE = orthochromatophilic erythroblasts; NS = not significant.

The online version of this article includes the following source data and figure supplement(s) for figure 4:

**Source data 1.** Source data for gene expression downstream of erythropoietin (EPO) in sorted bone marrow erythroblasts from wild type (WT), myelodysplastic syndrome (MDS), and DFP-treated MDS mice.

**Figure supplement 1.** Effect of DFP on erythroblast *Bcl11a* expression in WT mice.

**Figure supplement 1—source data 1.** Source data for B cell lymphoma (*Bcl11a*) in sorted bone marrow erythroblasts from wild type (WT) and DFP-treated WT mice.

**Figure supplement 2.** Nuances of signaling downstream of erythropoietin (EPO) in bone marrow erythroblasts from MDS and DFP-treated MDS mice.

**Figure supplement 2—source data 1.** Western blots with pSTAT5, STAT5, pAKT, and AKT antibody staining relative to actin in bone marrow erythroblast-enriched CD45 negative cells from myelodysplastic syndrome (MDS) and DFP-treated MDS mice.

**Figure supplement 2—source data 2.** Source data for quantification of signaling via STAT5 and AKT in bone marrow erythroblast-enriched CD45 negative cells from myelodysplastic syndrome (MDS) and DFP-treated MDS mice.

## Defective enucleation in MDS erythroblasts is normalized by DFP

Because increased EPO is implicated in defective enucleation (*Zhao et al., 2016a*), we also evaluate erythroblast enucleation in WT, MDS, and DFP-treated MDS mice. Our results demonstrate decreased enucleation in bone marrow erythroblasts from MDS relative to WT mice and return to normal expression levels in DFP-treated MDS mice (*Figure 5—figure supplement 2*). These findings are consistent with prior work which provides mechanistic evidence of an enucleation defect in MDS (*Zhao et al.,*

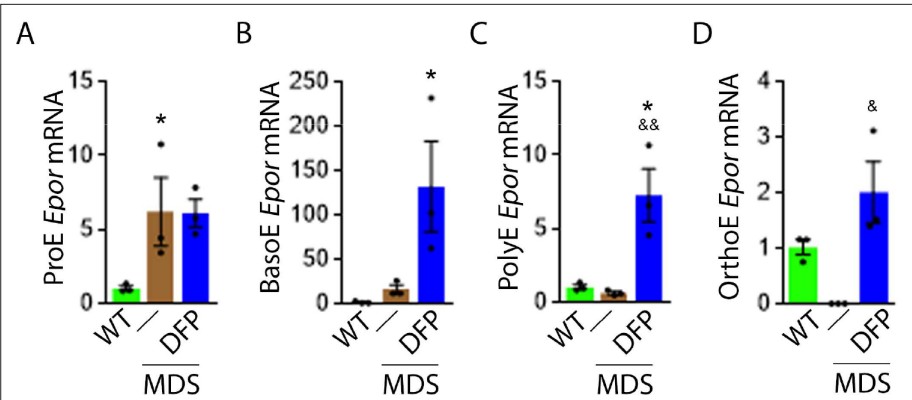

**Figure 5.** DFP increases *Epor* expression in MDS erythroblasts. DFP treatment in MDS mice increases *Epor* mRNA expression in sorted bone marrow following ProE (**A**), namely in BasoE (**B**), PolyE (**C**), and OrthoE (**D**) erythroblasts relative to untreated MDS or WT mice. *p<0.05 vs. WT; &p<0.05 vs. MDS; &&p<0.01 vs. MDS. Abbreviations: WT = wild type; MDS = myelodysplastic syndrome; DFP = deferiprone; Epor = erythropoietin receptor; ProE = pro-erythroblasts; BasoE = basophilic erythroblasts; PolyE = polychromatophilic erythroblasts; OrthoE = orthochromatophilic erythroblasts.

The online version of this article includes the following source data and figure supplement(s) for figure 5:

**Source data 1.** Source data for erythropoietin receptor (*Epor*) in sorted bone marrow erythroblasts from wild type (WT), myelodysplastic syndrome (MDS), and DFP-treated MDS mice.

**Figure supplement 1.** Effect of DFP on erythroblast *Epor* expression in WT mice.

**Figure supplement 1—source data 1.** Source data for erythropoietin receptor (*Epor*) in sorted bone marrow erythroblasts from wild type (WT) and DFP-treated WT mice.

**Figure supplement 2.** Defective enucleation in MDS erythroblasts is normalized by DFP.

**Figure supplement 2—source data 1.** Source data for flow analysis of enucleation in erythroblasts from wild type (WT), myelodysplastic syndrome (MDS), and DFP-treated MDS mice.

---

*2016b*). Our findings are also consistent with our previously published evidence demonstrating that manipulating iron metabolism in erythropoiesis leads to reversal of ineffective erythropoiesis is associated with normalization of the erythroblast enucleation defect in β-thalassemic mice, another model of ineffective erythropoiesis (*Li et al., 2017*).

## TFR1 but not TFR2 expression in MDS erythroblasts is normalized by DFP

Decreased iron in the bone marrow of DFP-treated MDS mice prompted us to evaluate whether specific mechanisms involved in iron sensing and trafficking could explain the beneficial effects of DFP on ineffective erythropoiesis in MDS mice. First, we demonstrate increased cell surface TFR1 on bone marrow erythroblasts from MDS relative to WT mice, normalized in DFP-treated MDS mice (*Figure 6A*). In contrast, cell surface TFR1 on bone marrow erythroblasts from DFP-treated WT mice is increased relative to WT mice (*Figure 6—figure supplement 1*). Because *Tfrc* expression in bone marrow is mainly EPO-mediated (*Chan et al., 1994*), we anticipate that erythroblast *Tfrc* expression is elevated in MDS as a consequence of high EPO, consequently decreased in DFP-treated relative to untreated MDS mice. We used sorted bone marrow to evaluate *Tfrc* expression in progressive stages of terminal erythropoiesis. First, *Tfrc* expression is increased in ProE and BasoE stages in MDS mice and remains elevated in DFP-treated MDS mice (*Figure 6B and C*). Next, *Tfrc* expression is normal and decreased in PolyE and OrthoE stages, respectively, in MDS relative to WT mice and increased to normal levels in DFP-treated relative to untreated MDS mice (*Figure 6D and E*). In contrast, *Tfrc* expression in bone marrow erythroblasts from DFP-treated WT mice is decreased relative to WT mice (*Figure 6—figure supplement 2*). These findings demonstrate that *Tfrc* expression correlates with *Epor* expression, especially in ProE stage, DFP in MDS erythroblasts leads to improvement in EPO-responsiveness not observed in WT mice, and erythroblast cell surface TFR1 for iron uptake during erythropoiesis is coordinated with EPO-responsiveness.

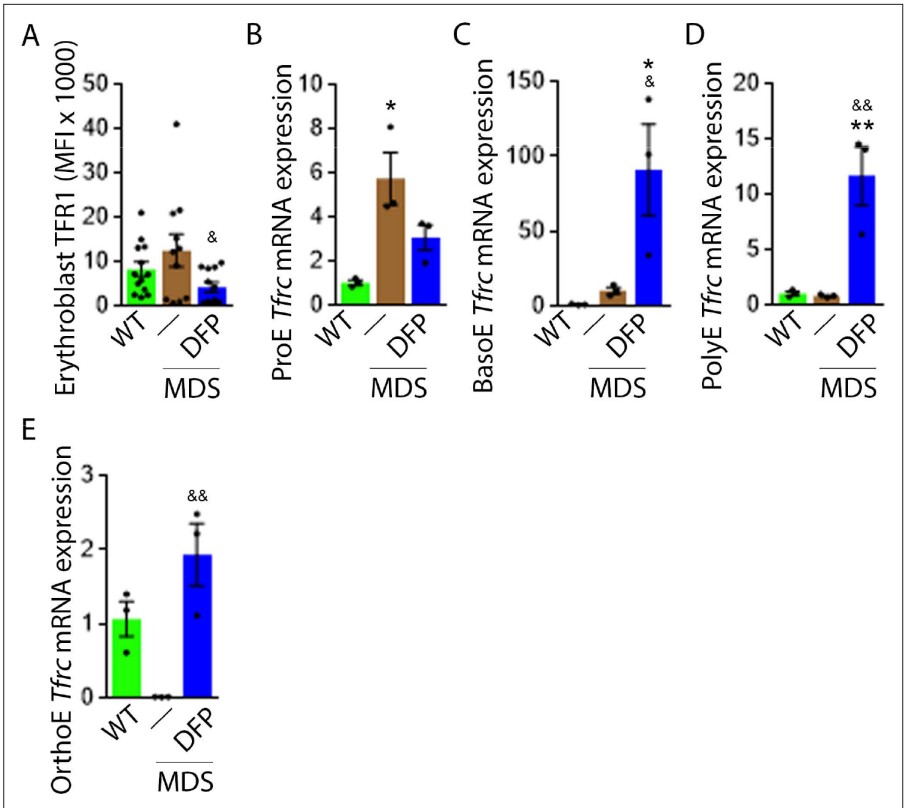

**Figure 6.** DFP normalized TFR1 expression in MDS erythroblasts. (**A**) Flow cytometry gating using TER119 and CD44 was used to delineate bone marrow erythroblasts. In these gated erythroblasts, we evaluate membrane TFR1 to demonstrate its decrease in response to DFP in MDS mice (n=7–13 mice/group) analyzed after 1 month of treatment. *Tfrc* mRNA expression is increased in sorted bone marrow ProE (**B**) from MDS mice and partially normalized in DFP-treated MDS mice. *Tfrc* mRNA expression is normal in sorted bone marrow BasoE (**C**) and PolyE (**D**), decreased in OrthoE (**E**) from MDS mice, and increased in DFP-treated relative to untreated MDS mice (n=15–18 mice/group). *p<0.05 vs. WT; **p<0.01 vs. WT; &p<0.05 vs. MDS; &&p<0.01 vs. MDS; Abbreviations: WT = wild type; MDS = myelodysplastic syndrome; DFP = deferiprone; TFR1=transferrin receptor 1; ProE = pro-erythroblasts; BasoE = basophilic erythroblasts; PolyE = polychromatophilic erythroblasts; OrthoE = orthochromatophilic erythroblasts.

The online version of this article includes the following source data and figure supplement(s) for figure 6:

**Source data 1.** Source data for flow analysis of cell surface TFR1 and *Tfrc* in sorted bone marrow erythroblasts from wild type (WT), myelodysplastic syndrome (MDS), and DFP-treated MDS mice.

**Figure supplement 1.** Effect of DFP on erythroblast surface TFR1 expression in WT mice.

**Figure supplement 1—source data 1.** Source data for flow analysis of cell surface transferrin receptor 1 (TFR1) on bone marrow erythroblasts from wild type (WT) and DFP-treated WT mice.

**Figure supplement 2.** Effect of DFP on erythroblast *Tfrc* expression in erythroblasts from WT mice.

**Figure supplement 2—source data 1.** Source data for *Tfrc* in sorted bone marrow erythroblasts from wild type (WT) and DFP-treated WT mice.

Next, we evaluate levels of TFR2 in light of its role in iron sensing and coordination of EPO-responsiveness with iron supply during erythropoiesis (*Ghaffari et al., 2006*; *Khalil et al., 2017*). Furthermore, TFR2 is under investigation as a potential therapeutic target in β-thalassemia, another disease of ineffective erythropoiesis (*Artuso et al., 2018*). We hypothesize that TFR2, given the proposed interaction with EPOR (*Forejtnikovà et al., 2010*), plays a central compensatory role in ineffective erythropoiesis. MDS mice exhibit higher erythroblast surface TFR2 expression specifically in ProE relative to WT, unchanged in DFP-treated MDS mice (*Figure 7A*). In addition, TFR2 expression is also borderline increased in bone marrow ProE erythroblasts, significantly higher in DFP-treated MDS relative to WT mice (*Figure 7B*); no significant differences are evident in BasoE, PolyE, and OrthoE

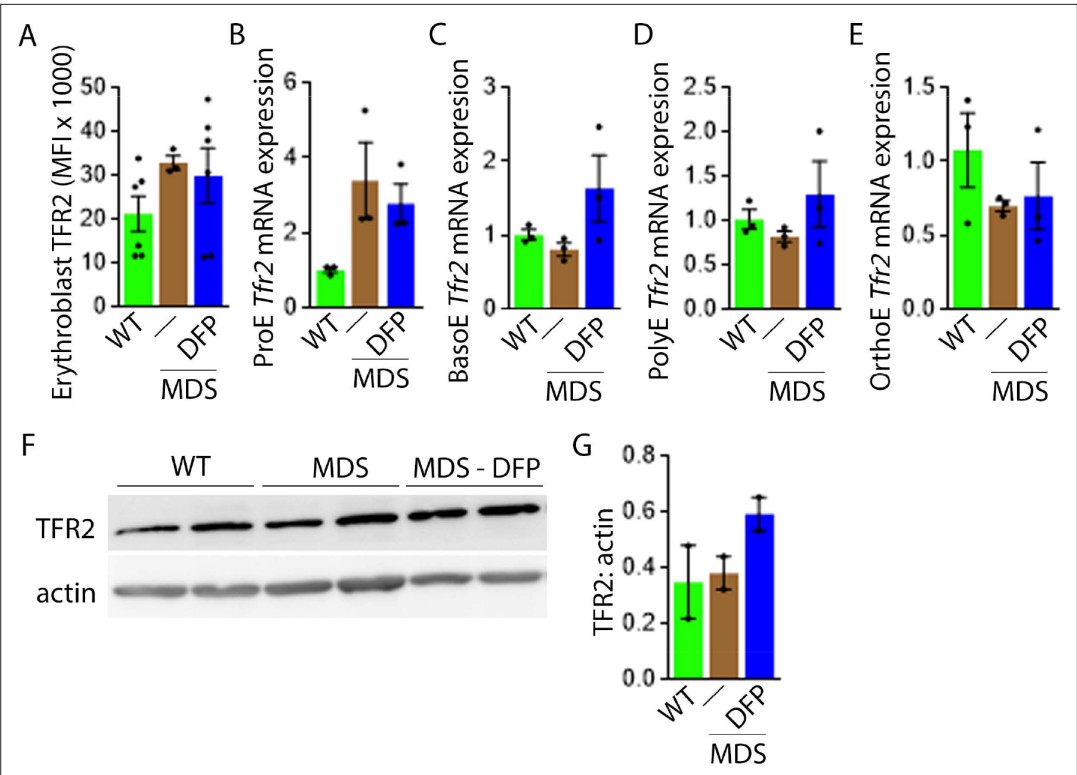

**Figure 7.** DFP does not normalize elevated TFR2 expression in MDS erythroblasts. (**A**) Flow cytometry gating using TER119 and CD44 was used to delineate bone marrow erythroblasts. In these gated erythroblasts, we evaluate membrane TFR2 which is unchanged in MDS and DFP-treated MDS mice (n=6–7 mice/group) analyzed after 1 month of treatment. (**B**) TFR2 mRNA expression is borderline increased in sorted bone marrow ProE from MDS relative to WT mice and remains significantly elevated in DFP-treated MDS relative to WT mice (n=15–18 mice/group). When compared with sorted bone marrow ProE, TFR2 expression in sorted bone marrow BasoE (**C**), PolyE (**D**), and OrthoE (**E**) is significantly suppressed in MDS relative to WT mice and remains suppressed in PolyE and OrthoE from DFP-treated MDS mice (n=15–18 mice/group). (**F**) Western blot demonstrating TFR2 protein concentration in bone marrow erythroblasts is not different between WT, MDS, and DFP-treated MDS mice, quantified in (**G**) (n=3 mice/sample). *p<0.05 vs. WT; **p<0.01 vs. WT; Abbreviations: WT = wild type; MDS = myelodysplastic syndrome; DFP = deferiprone; TFR2 = transferrin receptor 2; ProE = pro-erythroblasts; BasoE = basophilic erythroblasts; PolyE = polychromatophilic erythroblasts; OrthoE = orthochromatophilic erythroblasts.

The online version of this article includes the following source data and figure supplement(s) for figure 7:

**Source data 1.** Source data for flow analysis of cell surface transferrin receptor 2 (TFR2), TFR2 protein concentration in bone marrow erythroblast-enriched CD45 negative cells, and TFR2, in sorted bone marrow erythroblasts from wild type (WT), myelodysplastic syndrome(MDS), and DFP-treated MDS mice.

**Source data 2.** Western blots with transferrin receptor 2 (TFR2) antibody staining relative to actin in bone marrow erythroblast enriched CD45 negative cells from wild type (WT), myelodysplastic syndrome (MDS), and DFP-treated MDS mice.

**Figure supplement 1.** Erythroblast TFR2 mRNA expression in WT, MDS, and DFP-treated MDS mice.

**Figure supplement 1—source data 1.** Source data for transferrin receptor 2 (TFR2) in sorted bone marrow BasoE, PolyE, and OrthoE from wild type (WT), myelodysplastic syndrome (MDS), and DFP-treated MDS mice.

**Figure supplement 2.** Erythroblast *Scrib* mRNA expression in WT, MDS, and DFP-treated MDS mice.

**Figure supplement 2—source data 1.** Source data for Scribble (*Scrib*) in sorted bone marrow erythroblasts from wild type (WT), myelodysplastic syndrome (MDS), and DFP-treated MDS mice.

bone marrow erythroblasts between WT, MDS, and DFP-treated MDS mice (***Figure 7—figure supplement 1***). When compared with ProE, TFR2 expression in BasoE, PolyE, and OrthoE is significantly suppressed and remains suppressed in DFP-treated MDS mice (***Figure 7C***). Finally, no obvious differences in TFR2 protein concentration (***Figure 7D and E***) or *Scrib* mRNA expression (***Figure 7—figure***

*supplement 2*) are evident between bone marrow erythroblasts from WT, MDS, and DFP-treated MDS mice. These findings do not identify a DFP-specific modification of TFR2 mRNA, protein, or erythroblast surface localization or its action through changes in *Scrib* (*Khalil et al., 2018*). A mechanistic role for TFR2 in ineffective erythropoiesis in MDS remains to be fully elucidated.

## Altered iron trafficking in MDS erythroblasts is partially restored by DFP

We explored further the expression of other iron chaperones in MDS mouse bone marrow erythroblasts, hypothesizing that improved erythropoiesis in DFP-treated MDS mice is a consequence of not only decreased iron concentration but altered iron trafficking within erythroblasts. The cytosolic chaperone Poly(rC)-binding protein 1 (PCBP1) delivers iron to ferritin (*Leidgens et al., 2013*; *Ryu et al., 2017*) with evidence from *Pcbp1* knockout mice, with microcytosis and anemia, that iron delivery to ferritin is required for normal erythropoiesis (*Ryu et al., 2017*). In addition, PCBP2 is also required for ferritin complex formation (*Leidgens et al., 2013*). Conversely, an autophagic process to extract iron from the ferritin core is mediated by nuclear receptor coactivator 4 (NCOA4), a selective cargo receptor for autophagic ferritin turn-over, critical for regulation of intracellular iron availability (*Mancias et al., 2014*; *Dowdle et al., 2014*). In iron-replete states, PCBP1 and PCBP2 expression is enhanced while NCOA4 is targeted to the proteasome for degradation (*Mancias et al., 2015*; *Philpott, 2018*). Our results demonstrate that mRNA expression of both *Pcbp1* (*Figure 8A–C*) and *Pcbp2* (*Figure 8—figure supplement 1*) in sorted bone marrow ProE, BasoE, and PolyE is elevated in MDS mice and does not return to control WT levels in DFP-treated MDS mice. Conversely, mRNA expression of both *Pcbp1* (*Figure 8D*) and *Pcbp2* (*Figure 8—figure supplement 1*) in sorted bone marrow OrthoE is decreased in MDS relative to WT mice, normalized in DFP-treated MDS mice.

Furthermore, ProE mRNA expression of *Ncoa4* shows an increased trend in MDS mice and does not return to control WT levels in DFP-treated MDS mice (*Figure 8E*). While no differences in *Ncoa4* expression in bone marrow sorted BasoE and PolyE from WT, MDS, and DFP-treated MDS mice (*Figure 8F and G*), *Ncoa4* expression is suppressed in sorted bone marrow OrthoE in MDS mice, normalized in DFP-treated MDS mice (*Figure 8H*). Because of the role of NCOA4 in ferritinophagy and ferroptosis, we evaluate *Gpx4* expression in sorted bone marrow erythroblasts, demonstrating no differences between WT, MDS, and DFP-treated MDS mice (*Figure 8—figure supplement 2*). These findings are consistent with expectations that high levels of iron flux through ferritin, high rates of ferritin turnover, and high rates of iron transfer to the mitochondria require elevated NCOA4 and PCBP1/2 levels (*Mancias et al., 2015*) and provide preliminary evidence that movement of iron between sub-cellular compartments is altered in MDS erythroblasts, especially in early stages of terminal erythropoiesis, partially normalized by DFP.

## Increased expression of iron metabolism-related genes in MDS patient bone marrow stem and progenitor cells

Expression of iron metabolism-related genes is compared in bone marrow-derived CD34 + stem and progenitor cells from MDS patients (N=183) and healthy controls (N=17) as previously described (*Pellagatti et al., 2010*). As expected, *Tfrc*, *Epor*, *Gata1*, *Bcl2l1* (gene name for Bcl-Xl), and *Erfe* expression are significantly increased in MDS patients relative to controls (*Figure 9A–E*). In addition, while *Pcbp1* is unchanged and *Pcbp2* is borderline increased, *Ncoa4* is significantly increased in bone marrow stem and progenitor cells from MDS patients (*Figure 9F–H*), enabling increased ferritin degradation in MDS erythroblasts. Finally, TFR2 expression is also significantly increased in MDS patients relative to controls (*Figure 9I*), confirming our results in MDS mice. Whether changes in iron sensing and trafficking within erythroblasts contribute to MDS pathophysiology and ineffective erythropoiesis more broadly is unexplored. Our findings demonstrate that NHD13 mice recapitulate pathophysiological changes in iron sensing and trafficking in erythroid progenitors from MDS patients.

## Discussion

We show that MDS mice exhibit aberrant erythroblast iron trafficking and that iron chelation with DFP abrogates these changes to restore EPO-responsiveness. The effectiveness of DFP has previously been demonstrated in MDS patients (*Kersten et al., 1996*). These data raise the possibility that

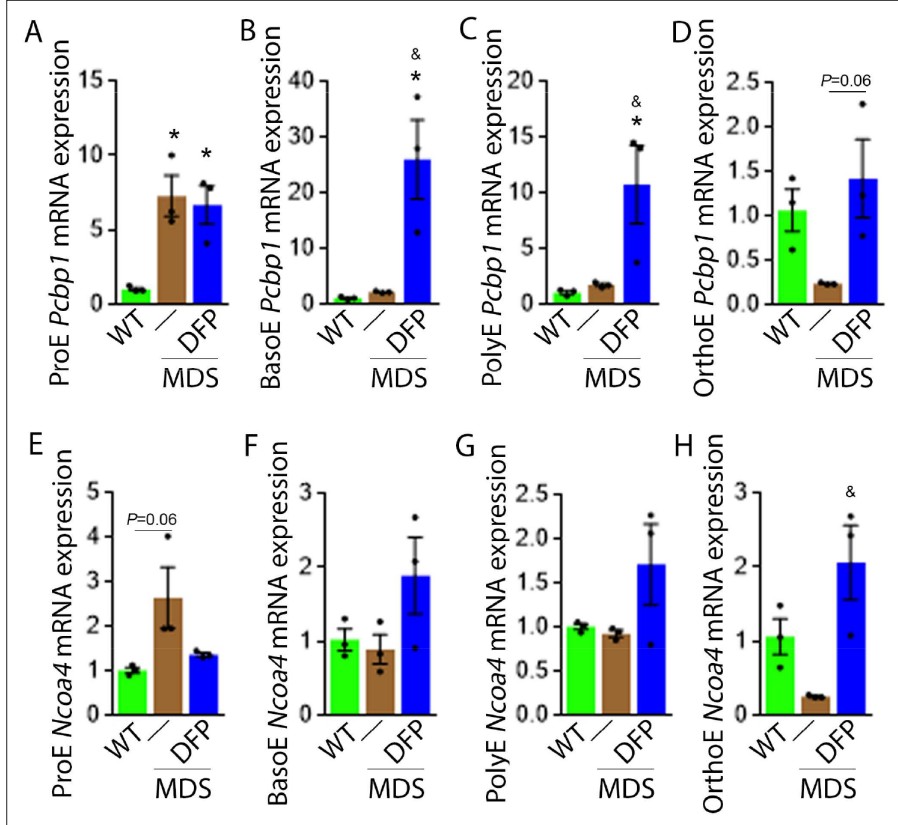

**Figure 8.** DFP alters the expression of iron chaperones in MDS erythroblasts. Sorted bone marrow *Pcbp1* mRNA expression is increased in ProE (**A**) from MDS relative to WT mice and remains high or is further elevated in ProE, BasoE (**B**), and PolyE (**C**) DFP-treated relative to untreated MDS mice. (**D**) Sorted bone marrow *Pcbp1* mRNA expression is unchanged in OrthoE from MDS relative to WT mice, borderline normalized in DFP-treated MDS mice. Sorted bone marrow mRNA expression of *Ncoa4* in ProE (**E**) is borderline increased in MDS, normalized in DFP-treated MDS mice. Sorted bone marrow mRNA expression of *Ncoa4* is unchanged in BasoE (**F**) and PolyE (**G**) from MDS or DFP-treated MDS relative to WT mice. Sorted bone marrow mRNA expression of *Ncoa4* in OrthoE (**H**) is unchanged in MDS and normalized in DFP-treated MDS mice (n=15–18 mice/group). *p<0.05 vs. WT; &p<0.05 vs. MDS; Abbreviations: WT = wild type; MDS = myelodysplastic syndrome; DFP = deferiprone; *Pcbp1*=Poly(rC)-binding protein 1; *Ncoa4*=nuclear receptor coactivator 4; ProE = pro-erythroblasts; BasoE = basophilic erythroblasts; PolyE = polychromatophilic erythroblasts; OrthoE = orthochromatophilic erythroblasts.

The online version of this article includes the following source data and figure supplement(s) for figure 8:

**Source data 1.** Source data for *Pcbp1* and *Ncoa4* in sorted bone marrow erythroblasts from wild type (WT), myelodysplastic syndrome (MDS), and DFP-treated MDS mice.

**Figure supplement 1.** Erythroblast *Pcbp2* mRNA expression in WT, MDS, and DFP-treated MDS mice.

**Figure supplement 1—source data 1.** Source data for *Pcbp2* in sorted bone marrow erythroblasts from wild type (WT), myelodysplastic syndrome (MDS), and DFP-treated MDS mice.

**Figure supplement 2.** Erythroblast *Gpx4* mRNA expression in WT, MDS, and DFP-treated MDS mice.

**Figure supplement 2—source data 1.** Source data for glutathione peroxidase 4 (*Gpx4*) in sorted bone marrow erythroblasts from wild type (WT), myelodysplastic syndrome (MDS), and DFP-treated MDS mice.

specifically targeting the iron sensing or iron trafficking machinery in erythroblasts may enable the amelioration of ineffective erythropoiesis in addition to iron overload in MDS patients.

RBCs, the highest concentration among all cell types in circulation, are a product of erythroid precursor differentiation and enucleation, requiring 80% of the circulating iron for Hb synthesis (*Muckenthaler et al., 2017*). Furthermore, how iron regulates erythropoiesis is incompletely understood and next to nothing is known about whether and how dysregulated iron metabolism contributes to the pathophysiology of ineffective erythropoiesis in MDS. The well-known, timely, abundant, and

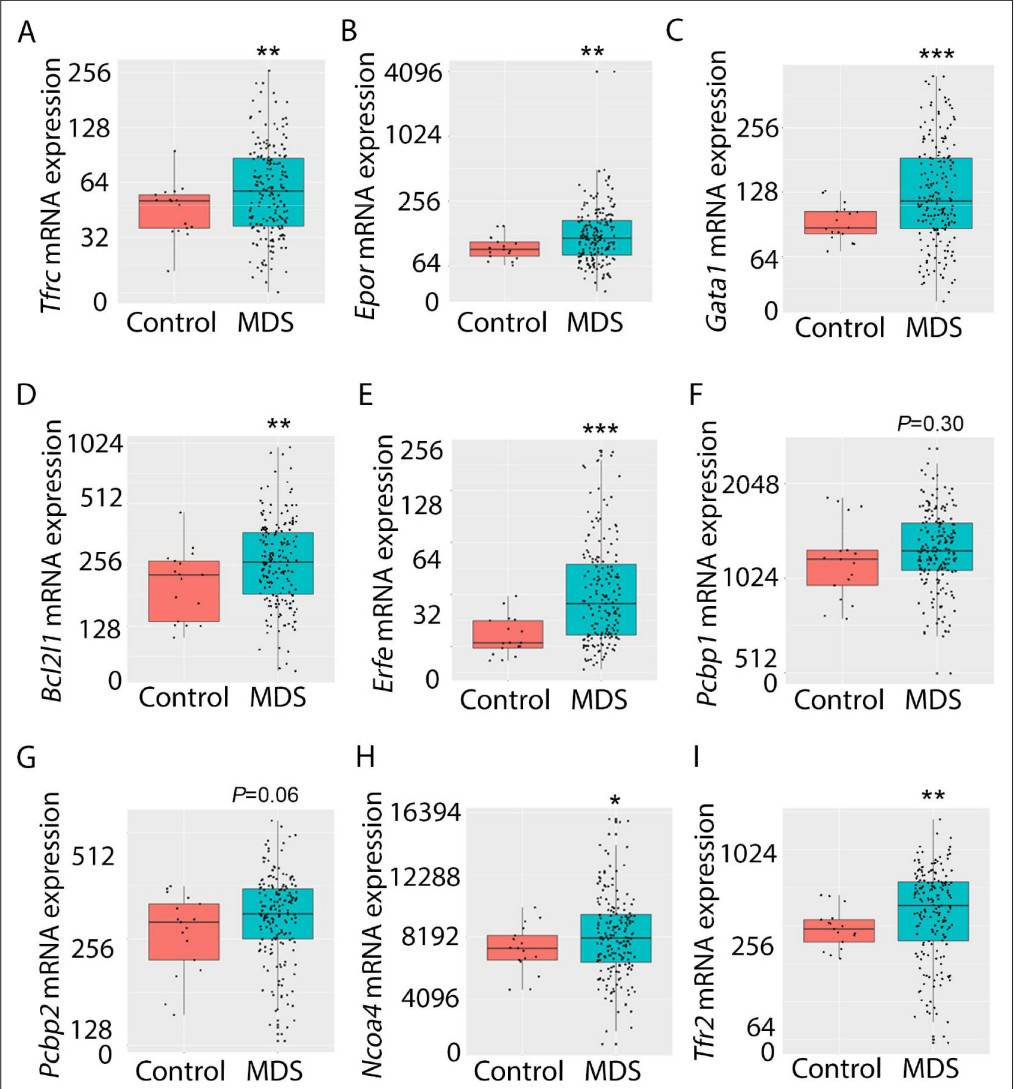

**Figure 9.** Increased expression of iron metabolism-related genes in MDS patient bone marrow stem and progenitor cells. Increased expression of *Tfrc* (**A**), *Epor* (**B**), *Gata1* (**C**), *Bcl2l1* (**D**), and *Erfe* (**E**) is expected and validates the database. No difference in *Pcbp1* (**F**), borderline increase in *Pcbp2* (**G**), and statistically significantly increase in *Ncoa4* (**H**) and TFR2 (**I**) are evident in MDS patients relative to controls, providing an important confirmation of the relevance of similar findings in MDS mice. *p<0.05, **p<0.01, ***p<0.0001 vs. control; MDS = myelodysplastic syndrome; *Tfrc* and TFR2 = transferrin receptor 1 and 2; Epor = erythropoietin receptor; *Bcl2l1* = B cell lymphoma 2-like protein 1 (gene name for Bcl-Xl); Erfe = erythroferrone; *Pcbp1* and *Pcbp2* = Poly(rC)-binding protein 1 and 2; *Ncoa4* nuclear receptor coactivator 4.

The online version of this article includes the following source data for figure 9:

**Source data 1.** Source data for iron metabolism-related genes in bone marrow stem and progenitor cells from myelodysplastic syndrome (MDS) patients vs. controls.

coordinated delivery of sufficient iron to erythroid precursors is accomplished via TFR1 to enable Hb production. TFR1 is both regulated by iron and enhanced by EPO-mediated signaling, evidence of the iron dependency in erythropoiesis. Furthermore, prior studies demonstrate that iron delivery to ferritin is absolutely required for normal erythropoiesis (*Ryu et al., 2017*) and cytosolic chaperones PCBP1 and PCBP2 were recently identified as central to ferritin iron delivery in erythroblasts (*Leidgens et al., 2013*; *Ryu et al., 2017*). In addition, the process of ferritinophagy has recently been described, as an autophagic process to extract iron from the ferritin core. NCOA4 is a selective cargo receptor for autophagic ferritin turn-over, critical for regulating intracellular iron availability for cellular

function (*Mancias et al., 2014*; *Dowdle et al., 2014*; *Mancias et al., 2015*; *Philpott, 2018*). Finally, TFR2 expression has recently been identified in erythroid precursors (*Nai et al., 2015*; *Khalil et al., 2017*; *Artuso et al., 2018*; *Forejtnikovà et al., 2010*; *Lee et al., 2012*; *Rishi et al., 2016*). TFR2 is functionally involved in erythroid differentiation (*Forejtnikovà et al., 2010*) via an interaction with EPOR (*Khalil et al., 2017*) to modulate EPO responsiveness and possibly in shuttling iron to the mitochondria, but conflicting data findings underscore our incomplete understanding of the downstream effect of TFR2 in erythropoiesis. Taken together, the movement of iron within erythroblasts is

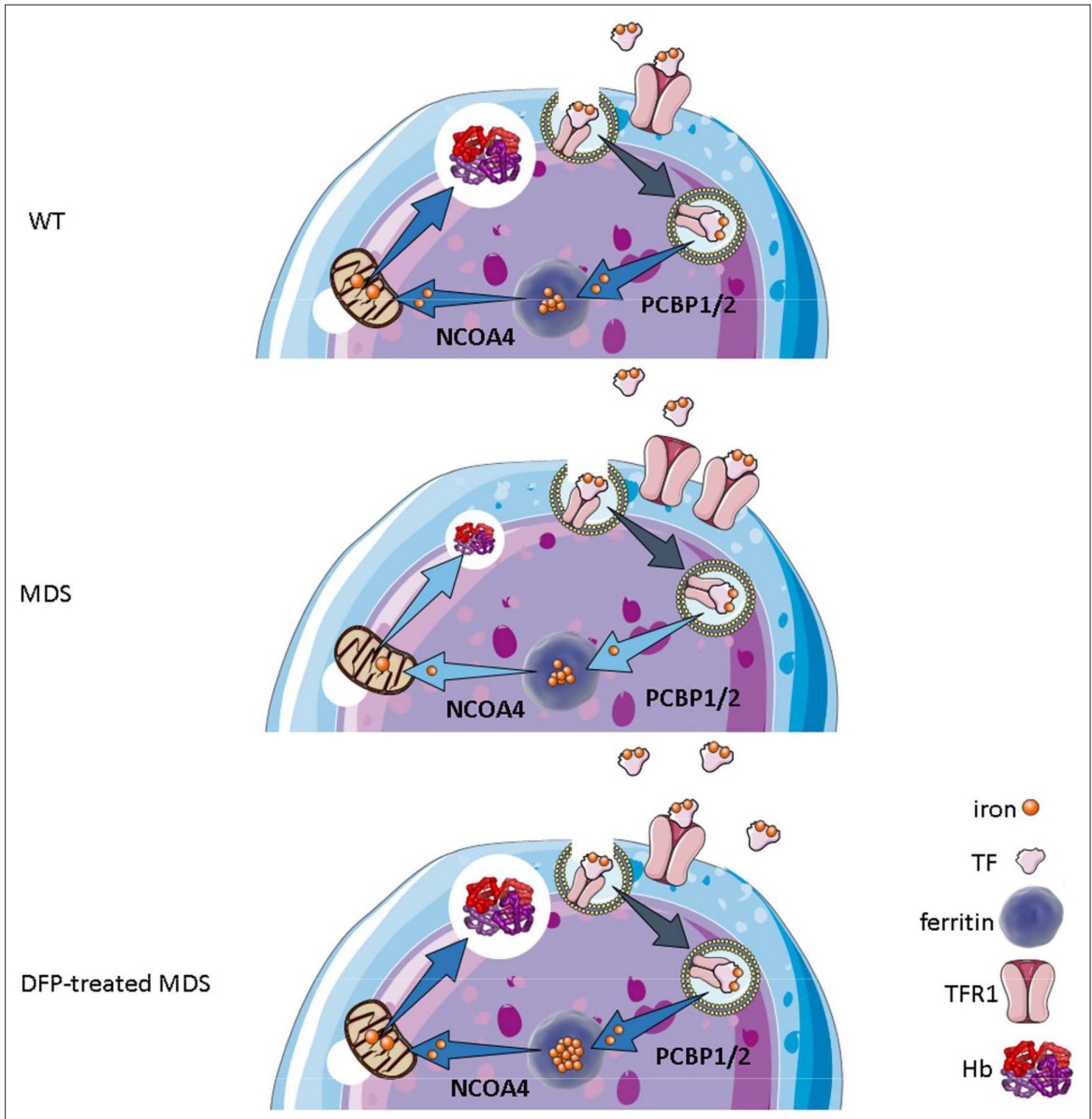

**Figure 10.** Putative function of erythroblast iron trafficking in health and in MDS before and after DFP treatment. Erythroblast iron uptake is mediated by TFR1 via endocytosis of clathrin-coated pits. In addition, erythroblast ferritin iron delivery is an obligatory step in erythropoiesis, and chaperones, i.e., PCBP1 and PCBP2, deliver iron to ferritin while NCOA4 enable iron extraction from ferritin, leading to iron utilization for hemoglobin synthesis in the mitochondria (**A**). In conditions of expanded erythropoiesis, such as MDS (**B**), more TFR1 results in increased iron uptake, resulting in decreased PCBP1 and NCOA4; yet no change in ferritin storage. Treatment with DFP restores erythroblast iron trafficking with decreased TFR1, increased PCBP1 and NCOA4, and increased ferritin concentration (**C**). We propose a model in which altered iron delivery to the mitochondria correlates with increased erythroblast survival and proliferation and decreased erythroid differentiation, causing ineffective erythropoiesis in MDS, ameliorated in response to DFP. Abbreviations: WT = wild type; MDS = myelodysplastic syndrome; DFP = deferiprone; TF = transferrin; TFR1 = transferrin receptor 1; PCBP1/2=Poly(rC)-binding protein 1 and 2; NCOA4=nuclear receptor coactivator 4; Hb = hemoglobin.

a complex multi-step process to handle a redox-active compound that is also required for the central task of erythropoiesis, i.e., Hb synthesis.

Our results demonstrate that iron trafficking is abnormal in MDS erythroblasts and restored in response to DFP (*Figure 10*). Specifically, increased TFR1 in MDS erythroblasts is expected to deliver more iron to erythroblast ferritin stores via increased PCBPs; more NCOA4 is expected to extract more iron from ferritin and more TFR2 to deliver more iron to mitochondria for Hb synthesis. Increased iron uptake by erythroblasts leads to a relatively normal amount of iron in the circulation despite systemic and parenchymal iron overload in MDS mice. Interestingly, erythroblast ferritin does not increase in MDS relative to WT mice; we speculate that this results from increased NCOA4-mediated iron release from ferritin and leads to an increase in iron's redox activity and higher erythroblast ROS. Our findings suggest that, although regulation of late-stage erythropoiesis remains incompletely understood, redox-active iron accumulation may lead to a feedback downregulation in iron trafficking genes during erythroblast differentiation, consequently resulting in ineffective erythropoiesis.

Differences in early and later stage erythroblast can also be seen when comparing iron trafficking genes in MDS mice with that in stem and progenitor cells from MDS patients. Patient samples correlate with early-stage erythroblasts in MDS mice in which *Tfrc*, *Epor*, TFR2, and NCOA4 expression is increased, providing validation that altered iron trafficking is a relevant pathophysiological component in MDS patients. In further support, AML patients for example exhibit increased bone marrow TFR2 expression (*Veneri et al., 2005*; *Kollia et al., 2003*), and expression in blasts correlates with serum ferritin and overall prognosis (*Nakamaki et al., 2004*). Based on these findings, we propose that targeting iron trafficking in erythroblasts may be a viable therapeutic strategy in iron-loading anemias, increasing erythroblast ferritin iron sequestration to protect later-stage erythropoiesis from redox-active effects of more labile forms of iron. For example, while NCOA4 is fundamental for iron supply during erythropoiesis, when iron is abundant, such as in MDS, other compensatory mechanisms are activated to prevent anemia (*Nai et al., 2021*), providing a rationale for targeting NCOA4 suppression to reverse ineffective erythropoiesis.

Lastly, although clearly, increased ROS levels result in cellular damage, emerging evidence suggests that ROS is required for normal hematopoiesis, influencing stem cell migration, differentiation, cell cycle status, and self-renewal (*Ludin et al., 2014*) such that hematopoietic stem cell self-renewal potential is associated with low ROS states while high ROS states are associated with differentiating hematopoietic stem cells. Further increased ROS leads to senescence and decreased ROS restores differentiation in those conditions (*Ludin et al., 2014*). In addition, ROS can activate JAK/STAT pathways (*Tibaldi et al., 2020*) and, therefore, EPO-EPOR mediated cell growth and survival. Recent evidence indicates that EPO and iron are required for ROS generation in erythroblasts, and that ROS are necessary for terminal erythropoiesis [44] while unchecked ROS accumulation results in anemia (*Friedman et al., 2001*; *Hebbel, 1990*; *Kong et al., 2004*; *Lee et al., 2004*; *Neumann et al., 2003*). Taken together, ROS generation is both critical and potentially toxic, requiring significant coordination during erythropoiesis with particular importance for mitigating increased ROS in conditions of ineffective erythropoiesis as in MDS. However, whether increased ROS resulting in apoptosis as the cause of ineffective erythropoiesis has never been definitely confirmed. We and others previously demonstrated that EPO downstream mechanisms are potent anti-apoptotic compensatory mechanisms and that reversal of apoptosis does not ameliorate ineffective erythropoiesis in β-thalassemia (*Feola et al., 2021*; *Menon et al., 2022*). Our current results demonstrate borderline increased erythroblast apoptosis while ROS is significantly enhanced in MDS mice and no change in apoptosis while ROS is significantly decreased in DFP-treated MDS mice. These results uncouple apoptosis from ROS as the underlying cause of ineffective erythropoiesis.

In conclusion, despite FDA approval of iron chelators for use in iron-overloaded MDS patients, they remain underutilized due to a lack of biological insights into the deleterious effects of iron overload on disease pathophysiology in MDS. Our results demonstrate that DFP partially ameliorates ineffective erythropoiesis in MDS mice not only by reducing systemic iron overload, but also by altering iron trafficking within erythroblast and the sensitivity of erythroblasts to EPO, enhancing erythroblast differentiation. We further identify similar alterations in iron trafficking genes also in MDS patient bone marrow samples, validating our findings in MDS mice and anticipating a potential for translating these results. An important caveat to consider is whether a mouse model of transfusion-independent MDS such as NHD13 mice applies also to transfusion-dependent patients. The main focus of the current

work is to evaluate the effect of DFP on ineffective erythropoiesis. As a consequence, using transfused NHD13 mice is impossible as transfusion itself suppresses the very ineffective erythropoiesis we aim to study.

In addition, with an eye toward its translation, the use of DFP has been associated with the occurrence of severe neutropenia/agranulocytosis defined as an absolute neutrophil count below $0.5 \times 10^9$ cells/L. A recent report from the US safety registry of DFP indicates a potentially higher incidence of agranulocytosis in 115 patients with MDS, all resolved with dose interruption (*Zeidan et al., 2022*). The report concludes that the safety profile of DFP in MDS is potentially consistent with that of other iron-loading anemias with no new safety concerns or unexpected adverse events. However, it remains to be determined whether agranulocytosis occurs at a higher frequency than with other iron chelating agents in MDS, is a specific consequence unique to the patients' primary disease, or resulted from the relatively small sample size in this registry. Thus, although additional evidence is needed to ascertain the mechanism(s) and clinical relevance of DFP-induced agranulocytosis in MDS patients in anticipation of its translation, no change in white blood cell count was observed in mice and potential benefits may ultimately be found to outweigh the risks. Taken together, the main purpose of the current study is to provide a rationale for targeting erythroblast-specific iron sensing and trafficking to ameliorate ineffective erythropoiesis in MDS either with already available (e.g. DFP) or novel therapies currently in development for diseases of ineffective erythropoiesis.

## Materials and methods
### Mice and treatment
C57BL/6 (WT) and C57BL/6-Tg(Vav1-NUP98/HOXD13)G2Apla/J (NHD13) mice (*Lin et al., 2005*) were originally purchased from Jackson Laboratories (Bar Harbor, ME, USA). For simplicity, NHD13 mice are designated as 'MDS mice' throughout the manuscript. All mice were bred and housed in the animal facility under the Association for Assessment and Accreditation of Laboratory Animal Care guidelines. Experimental protocols were approved by the Institutional Animal Care and Use Committee. This well-established mouse model has been shown to recapitulate all key findings in human MDS, including blood cell dysplasia, peripheral blood cytopenias, ineffective hematopoiesis prior to transformation to acute leukemia, and a subset of mice progressing to acute leukemia at 14 months (*Lin et al., 2005*; *Slape et al., 2008*). NHD13 mice on a C57BL/6 background were previously found to be clinically appropriate as an MDS model until at least 7 months (*Lin et al., 2005*; *Slape et al., 2008*). As a consequence, we used age and gender-matched 5-month-old mice, at least five mice per group, treated with deferiprone (DFP; trade name Ferriprox; chemical name 3-hydroxy-1,2-dimethylpyridin -4-one) at a concentration of 1.25 mg/mL in the drinking water for 4 weeks. DFP is an orally active iron chelator that binds iron in a 3:1 (DFP:iron) complex and undergoes renal clearance. Mice were euthanized for analysis at 6 months of age, as mice were analyzed previously to study the effects of ineffective erythropoiesis (*Suragani et al., 2014*). Therefore, all endpoints of interest were analyzed to compare DFP-treated MDS mice after 1 month of treatment with untreated MDS mice and WT controls.

### Peripheral blood analyses
Mice peripheral blood cell counting was analyzed by ProCyte Dx Hematology Analyzer. Serum mouse EPO (Quantikine, R&D Systems) was measured by enzyme-linked immunosorbent assay (ELISA) according to the manufacturer's instructions. Integra 800 Automated Clinical Analyzer (Roche Diagnostics) was used to measure serum iron to transferrin iron-binding capacity (TIBC); serum transferrin saturation was measured as a ratio of serum iron to TIBC. Serum DFP and its metabolite DFP-glucuronide (DFP-G) concentrations were measured using liquid chromatography–mass spectrometry, validated in accordance with the FDA Bioanalytical Method Validation Guidance. The lower limit of quantifiable values for the method was 0.10 µM for DFP and 0.050 µM for DFP-Glu in mouse serum.

### Histology and immunohistochemistry
Immunohistochemical staining was performed using anti-TER119 antibodies (eBioscience, San Diego, CA), and counterstained with hematoxylin. Images were acquired on a Zeiss Axioskop2 microscope with an AxioCamHRC camera using Plan-Neofluar objectives 20x/0.5 and Axiovision software.

### Non-heme iron spectrophotometry

Quantification was performed via the Torrance and Bothwell method (*Torrance and Bothwell, 1980*). Briefly, specimens were digested overnight in an acid solution at 65 °C. A mixture of chromogen solution with acid extraction was incubated at room temperature for 10 min and the absorbance was measured at 540 nm by spectrophotometer (CLARIOstar plate reader, BMG Labtech).

### Quantitative real-time PCR

We prepared RNA from sorted bone marrow and liver samples using the RNeasy Kit (Qiagen) according to the manufacturer's instructions. We synthesized cDNA using the High-Capacity cDNA Reverse Transcription Kit (Thermo Fisher). Primers are listed in *Supplementary file 1a*. qPCR was conducted by iQ SYBRGreen Supermix using the BioRad CFX96 Real-Time PCR Thermal Cycler. Target gene mRNA concentration was normalized to *Gapdh*.

### Western immunoblotting

Beads sorted CD45 negative bone marrow or liver cells were lysed in ice-cold SDS page lysis buffer (2% SDS, 50 mM Tris-HCl, pH 7.4, 10 mM EDTA) with protease and phosphatase inhibitors. Twenty mg of heat–denatured protein was loaded onto a 10% gel, run, and transferred onto a 0.4 mm nitrocellulose membrane (Thermo Scientific). After blocking with 5% BSA in Tris–buffered saline with 1% Tween-20, the membranes were incubated with primary antibodies to signaling proteins (*Supplementary file 1b*) overnight at 4 °C, washed, and incubated with the corresponding HRP–conjugated secondary antibodies at room temperature. Proteins were visualized using the ImageQuant LAS 4010 and quantified using Image J.

### Flow cytometric analysis and sorting

Bone marrow cells were processed as described previously (*Liu et al., 2013*) with minor modifications. Briefly, the cells were mechanically dissociated, blocked with rat anti–mouse CD16/CD32 (Fcγ III/II Receptor), incubated with anti-CD45 magnetic beads (Mylteni), and underwent magnetic separation using LS columns according to the manufacturer's instructions (Miltenyi Biotec). Erythroid lineage-enriched CD45 negative cells were collected for further staining. Non-erythroid and necrotic cells were excluded using anti-CD45 (BD Pharmigen), anti-CD11b, and anti-Gr1 (APC-Cy7) (Tonbo, Biosciences) antibodies. Cells were incubated with anti-mouse TER119 PE-Cy7 (BioLegend) and CD44-APC (Tonbo, Biosciences) to identify and delineate progressive stages of erythroblast differentiation (*Figure 3—figure supplement 6*). Once erythroblasts were delineated by TER119, CD44, and forward scatter, CD71-PE (BioLegend) was used to evaluate changes in erythroblast membrane TFR1. ROS quantification in erythroblasts was performed using immunostaining for ROS (Invitrogen) as per the manufacturer's instructions. To evaluate apoptosis, cells were stained for activated caspase 3/7 kit (Invitrogen); 7-amino-actinomycin D (7AAD, BD Pharmingen) was added to exclude dead cells. Cells were analyzed within 1 hr of staining using BD FACSDiva Version 6.1.2 software on a FACSCanto flow cytometer (Becton Dickinson). The gating strategy was as previously described (*Liu et al., 2013*). Erythroid differentiation was quantified by analyzing the fraction of each stage of terminal erythropoiesis relative to all erythroblasts in each bone marrow sample. In addition, to individually evaluate gene expression in erythroblasts at different maturation stages, bone marrow cells underwent sorting on a BD FACSAria III (BD Biosciences). Finally, enucleation was assessed as described previously (*An and Chen, 2018*).

### Gene expression in MDS patient database

Gene expression data from 183 MDS CD34 + samples and 17 controls were obtained from GEO (GSE19429) as previously described (*Pellagatti et al., 2010*).

### Statistical analyses

All data are reported as mean ± standard error of the mean (SEM). We performed analysis for statistically significant differences using one-way ANOVA with multiple comparisons and Tukey post-test analyses for the mouse samples where more than two groups are compared and two-tailed t-test in all other circumstances.

## Acknowledgements

We sincerely appreciate the perpetual energy and devotion of the late Professor Eliezer Rachmilewitz to the importance of understanding and treating iron overload in MDS and dedicate this manuscript to his memory. We are grateful for funding from CSL Behring (King of Prussia, PA) (to Y.Z.G.) and the Italian Government Fellowship (to M.F.) for support of early work that served as the foundation of this study and ApoPharma (Chiesi) for supporting the experiments presented here.

## Additional information

### Competing interests

Yelena Ginzburg: Reviewing editor, *eLife.* The other authors declare that no competing interests exist.

### Funding

| Funder | Grant reference number | Author |
| --- | --- | --- |
| Chiesi USA | ApoPharma Grant | Yelena Ginzburg |

The funders had no role in study design, data collection and interpretation, or the decision to submit the work for publication.

### Author contributions

Wenbin An, Data curation, Formal analysis, Investigation, Methodology, Writing - original draft, Writing – review and editing; Maria Feola, Maayan Levy, Data curation, Investigation, Methodology, Writing – review and editing; Srinivas Aluri, Resources, Data curation, Formal analysis, Investigation, Methodology, Writing – review and editing; Marc Ruiz-Martinez, Data curation, Methodology, Writing – review and editing; Ashwin Sridharan, Data curation, Formal analysis, Methodology, Writing – review and editing; Eitan Fibach, Conceptualization, Investigation, Writing – review and editing; Xiaofan Zhu, Resources, Data curation, Writing – review and editing; Amit Verma, Conceptualization, Resources, Data curation, Formal analysis, Supervision, Funding acquisition, Investigation, Project administration, Writing – review and editing; Yelena Ginzburg, Conceptualization, Resources, Data curation, Formal analysis, Supervision, Funding acquisition, Writing - original draft, Writing – review and editing

### Author ORCIDs

Amit Verma https://orcid.org/0000-0002-5408-1673
Yelena Ginzburg http://orcid.org/0000-0002-3496-3783

### Ethics

This study was performed in strict accordance with the recommendations in the Guide for the Care and Use of Laboratory Animals of the National Institutes of Health. All of the animals were handled according to approved institutional animal care and use committee (IACUC) protocols (#16-0143) of the Icahn School of Medicine at Mount Sinai. All surgery was performed under isoflurane anesthesia, and every effort was made to minimize suffering.

### Decision letter and Author response

Decision letter https://doi.org/10.7554/eLife.83103.sa1
Author response https://doi.org/10.7554/eLife.83103.sa2

## Additional files

### Supplementary files

• Supplementary file 1. Tables of reagents used for PCR and western blot experiments. (a) Table of primers. (b) Table of antibodies.

### Data availability

All data generated or analyzed during this study are included in the manuscript and supporting file. Source data files have been provided for Figures 1–9.

The following previously published dataset was used:

| Author(s) | Year | Dataset title | Dataset URL | Database and Identifier |
|---|---|---|---|---|
| Pellagatti A, Boultwood J, Wainscoat JS | 2010 | Expression data from bone marrow CD34+ cells of MDS patients and healthy controls | https://www.ncbi.nlm.nih.gov/geo/query/acc.cgi?acc=GSE19429 | NCBI Gene Expression Omnibus, GSE19429 |

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
