## [Editor Report]

This study presents a valuable finding on the preclinical studies of chelating agents deferiprone (DFP) as potential therapeutics in Myelodysplastic Syndrome (MDS). The evidence supporting the claims of the authors is solid. The work will be of interest to clinicians or biologists working on MDS.

---

## [Decision Letter]

**Decision letter after peer review:**

Thank you for submitting your article "Iron Chelation Improves Ineffective Erythropoiesis and Iron Overload in Myelodysplastic Syndrome Mice" for consideration by *eLife*. Your article has been reviewed by 3 peer reviewers, including Yongliang Yang as Reviewing Editor and Reviewer #1, and the evaluation has been overseen by Caigang Liu as the Senior Editor.

Essential revisions:

1) The authors should provide solid evidence and elaborate on the choice of mouse model. The NUP98-HOXD13 mouse model of MDS recapitulates spontaneous (non-transfusion related) iron overload seen in some subtypes of MDS.

2) The authors should clarify the rationale for the choice of deferiprone as the iron chelator.

3) There are also substantial technical concerns and some aspects that need to be updated to current standards.

4) It would be important to present representative gating from all 3 animal groups: WT, MDS, and MDS+DFP mice.

5) In the Statistical analysis section, the authors state that all comparisons were done with 2-tailed student paired t test, which would not be appropriate for comparisons being made between independent animals groups.

6) All the referees raised concerns about Figure 10. The schematic diagram is rather difficult to follow and therefore should be amended.

*Reviewer #1 (Recommendations for the authors):*

Some specific points are as follows,

1. The authors should elaborate in a separate paragraph in the Discussion section to justify their choice of NUP98-HOXD13 mice as representative of low-risk MDS for their experiments. In particular, the authors should carefully explain and comment such that the reader could understand the appropriateness and preclinical potential of NUP98-HOXD13 mice model for the study of low-risk MDS. This is crucial for the reader to fully appreciate the values and merits of their manuscript.

2. Agranulocytosis was implicated as side effects of DFP treatment for transfusional iron overload in clinical practice. The authors need to elaborate in a paragraph to project and discuss the possible adverse events of DFP chelation therapy for low-risk MDS patients should it be approved.

3. Table 1, caption, "Characteristics of NHD13 mice are consistent with elevated MCV anemia with expanded hematopoiesis and iron overload which is appropriate as a mouse model of MDS". There exists grammar mistakes in this caption and therefore it is difficult to understand. This should be fixed.

4. Figure 10, the schematic diagram is rather difficult to follow and understand for the readers. The authors should try to highlight the key differences under three conditions. Some details (font size et al.) should be polished as well.

*Reviewer #2 (Recommendations for the authors):*

There are also substantial technical concerns and some aspects that need to be updated to current standards (the manuscript appears to have been written a long time ago):

1. The rationale for the choice of deferiprone as the iron chelator is weak, since most of the clinical data were reported with deferasirox. It would be better to do an initial comparison of the two agents to document if they affect anemia and ineffective erythropoiesis differentially (along the lines of Figure 2 and 3E). This would put the DFP data in the paper in a more appropriate context.

2. The graph style needs to be updated to change bar graphs to scatterplots with error bars. This also allows display of mouse sex using different symbols (instead of “data not shown”).

3. Supplementary figures 3A and 3B use only 2 mice? Which band is FTH?

4. Supplementary figure 4 is very poor quality and cannot be used for quantification.

5. Supplementary figure 9 does not show visible changes in pStat5. With n = 3 mice the changes would have to be large to be significant.

6. Erfe is the approved gene symbol for erythroferrone https://www.genenames.org/data/gene-symbol-report/#!/hgnc_id/HGNC:26727

7. Figure 7C: Why was Figure 7C normalized to ProE? Supplementary Figure 13 seems more informative about the lack of differences between WT, MDS and MDS-DFP.

8. Figure 10 was not helpful to my understanding of the proposed model in the context of the data in this manuscript. For example, if TfR2 expression does not change between the groups per previous results, what is the evidence that it differentially affects EpoR signaling in this model?

*Reviewer #3 (Recommendations for the authors):*

Below are some suggestions that I believe would strengthen the science and its presentation.

1. In certain areas of the Results, the authors appear to overstate their conclusions. For example, on p. 8, the authors state that "… reversal of ineffective erythropoiesis… leads to restored Hamp responsiveness." While the authors show a correlation between improved erythropoiesis and improved Hamp responsiveness to iron stores, the wording here should be modified, as the authors do not prove causation.

2. Statistical analysis: Paired t tests may not be appropriate for those comparisons that do not involve repeated measures on the same groups. Consider an unpaired T test, or preferably a one-way ANOVA with an appropriate posthoc test. When the same groups are analyzed for multiple parameters (e.g. comparisons of different stages of terminal erythropoiesis in WT, MDS, and DFP-MDS groups), consider an ANOVA test with repeated measures, followed by an appropriate post-hoc test.

3. Figure 1: In addition to showing the non-heme iron content of liver and spleen, it would be valuable to present the histological distribution of iron within these organs.

---

## [Author Response]

Essential revisions:1) The authors should provide solid evidence and elaborate on the choice of mouse model. The NUP98-HOXD13 mouse model of MDS recapitulates spontaneous (non-transfusion related) iron overload seen in some subtypes of MDS.

We appreciate this request for clarification. We use NUP98-HOXD13 transgenic (NHD13) mice for several reasons: (1) NHD13 mice were generated by transgenic expression of the NUP98/ HOXD13 fusion protein found in human MDS and acute myeloid leukemia, making them mechanistically meaningful as a model of MDS. (2) NHD13 mice are characterized by abortive erythroblast maturation and consequent ineffective erythropoiesis, consistent with MDS patients and of specific interest for our purposes, namely searching for a mechanistic understanding of ineffective erythropoiesis to target its amelioration. (3) NHD13 mice were successfully used in pre-clinical investigation of an activin receptor II ligand trap, reversing ineffective erythropoiesis, work that provided valuable pre-clinical proof-of-principle, leading to successful clinical trials and now FDA approval of this agent for MDS patients. Taken together, NHD13 mice are a well-established in vivo model of MDS. Given the dearth of better alternative in vivo models of MDS, these experiments lead the field in exploring the effect of DFP on ineffective erythropoiesis in MDS.

We interpret the reviewer’s inquiry specifically around whether transfusion-independent iron overload mouse models can be used to also shed light on iron overload in transfusiondependent MDS. We are, however, not looking to demonstrate DFP-mediated reversal of systemic iron overload at all; we show the expected effects of DFP on reversal of systemic iron overload only as validation, using this known effect as an expected outcome (Figure 1). The main focus of the current work is on the effects of DFP on ineffective erythropoiesis. As a consequence, we cannot use transfused NHD13 mice as transfusion itself effectively suppresses ineffective erythropoiesis, preventing its evaluation in response to DFP.

Toward addressing this comment, we have expanded our description of these mice in the Introduction and added text to the Discussion regarding the use of this mouse model in the revised manuscript.

2) The authors should clarify the rationale for the choice of deferiprone as the iron chelator.

Thank you to the editor and reviewers for this request for clarification. The rationale for choosing DFP as the iron chelator for these studies involves several known elements: 1) DFP has a lower iron binding affinity relative to transferrin (Sohn et al., 2008); 2) DFP results in iron transfer from parenchymal cells to increase serum iron, transferrin saturation, and hepcidin responsiveness to iron (Figure 1); and 3) increased hepcidin responsiveness prevents further iron absorption and recycling (Schmidt et al., 2015; Casu et al., 2016). On the foundation of this knowledge, we hypothesize that in addition to decreasing progression of systemic iron overload, DFP could also ameliorate ineffective erythropoiesis in MDS. We anticipate that the specific mechanism is reliant on the ability of DFP to donate iron to transferrin, altering the balance between different transferrin moieties in circulation, resulting in altered iron trafficking within erythroblasts and enhancing erythroblast differentiation.

To clarify our rationale for choosing DFP, we have expanded this part of the Introduction in the revised manuscript.

3) There are also substantial technical concerns and some aspects that need to be updated to current standards.

We anticipate that this general comment is in reference to the specific technical elements raised by Reviewer #2. These have been addressed (see below).

4) It would be important to present representative gating from all 3 animal groups: WT, MDS, and MDS+DFP mice.

We appreciate this comment and have added representative gating for all 3 groups to Supplementary Figure 17 (new Figure 3 —figure supplement 6).

5) In the Statistical analysis section, the authors state that all comparisons were done with 2-tailed student paired t test, which would not be appropriate for comparisons being made between independent animals groups.

We appreciate this comment and have reanalyzed all revised mouse data using one-way ANOVA with multiple comparisons and Tukey post-test analyses when more than 2 groups were compared. This has been edited in the Methods section in the revised manuscript.

6) All the referees raised concerns about Figure 10. The schematic diagram is rather difficult to follow and therefore should be amended.

We have modified Figure 10 to simplify and increase clarity in response to the reviewers’ comments, highlighting only the differences in erythroblast iron trafficking between control, MDS, and DFP-treated MDS conditions as the reviewers suggest.

Reviewer #1 (Recommendations for the authors):Some specific points are as follows,1. The authors should elaborate in a separate paragraph in the Discussion section to justify their choice of NUP98-HOXD13 mice as representative of low-risk MDS for their experiments. In particular, the authors should carefully explain and comment such that the reader could understand the appropriateness and preclinical potential of NUP98-HOXD13 mice model for the study of low-risk MDS. This is crucial for the reader to fully appreciate the values and merits of their manuscript.

We appreciate this request for clarification. We use NUP98-HOXD13 transgenic (NHD13) mice for several reasons: 1) NHD13 mice were generated by transgenic expression of the NUP98/ HOXD13 fusion protein found in human MDS and acute myeloid leukemia, making them mechanistically meaningful as a model of MDS. 2) NHD13 mice are characterized by abortive erythroblast maturation and consequent ineffective erythropoiesis, consistent with MDS patients and of specific interest for our purposes, namely searching for a mechanistic understanding of ineffective erythropoiesis to target its amelioration. 3) NHD13 mice were successfully used in pre-clinical investigation of an activin receptor II ligand trap, reversing ineffective erythropoiesis, work that provided valuable pre-clinical proof-of-principle, leading to successful clinical trials and now FDA approval of this agent for MDS patients. Taken together, NHD13 mice are a well-established in vivo model of MDS. Given the dearth of better alternative in vivo models of MDS, these experiments lead the field in exploring the effect of DFP on ineffective erythropoiesis in MDS.

We interpret the reviewer’s inquiry specifically around whether transfusion-independent iron overload mouse models can be used to also shed light on iron overload in transfusion dependent MDS. We are, however, not looking to demonstrate DFP-mediated reversal of systemic iron overload at all; we show the expected effects of DFP on reversal of systemic iron overload only as validation, using this known effect as an expected outcome (Figure 1). The main focus of the current work is on the effects of DFP on ineffective erythropoiesis. As a consequence, we cannot use transfused NHD13 mice as transfusion itself effectively suppresses ineffective erythropoiesis, preventing its evaluation in response to DFP.

Toward addressing this comment, we have expanded our description of these mice in the Introduction and added text to the Discussion regarding the use of this mouse model in the revised manuscript.

2. Agranulocytosis was implicated as side effects of DFP treatment for transfusional iron overload in clinical practice. The authors need to elaborate in a paragraph to project and discuss the possible adverse events of DFP chelation therapy for low-risk MDS patients should it be approved.

While the purpose of the current work was to provide a rationale to specifically target erythroblast-specific iron sensing and trafficking which here was accomplished by DFP, we appreciate this forward-looking comment and have added more data about agranulocytosis to the Discussion section of the revised manuscript:

“In addition, with an eye toward its translation, the use of DFP has been associated with the occurrence of severe neutropenia/agranulocytosis defined as an absolute neutrophil count below 0.5 x 10^9^ cells/L. A recent report from the US safety registry of DFP indicates a potentially higher incidence of agranulocytosis in 115 patients with MDS, all resolved with dose interruption (Zeidan et al., 2022). The report concludes that the safety profile of DFP in MDS is potentially consistent with that of other iron-loading anemias with no new safety concerns or unexpected adverse events. However, it remains to be determined whether agranulocytosis occurs at higher frequency than with other iron chelating agents in MDS, is a specific consequence unique to the patients’ primary disease, or resulted from the relatively small sample size in this registry. Thus, although additional evidence is needed to ascertain the mechanism(s) and clinical relevance of DFP-induced agranulocytosis in MDS patients in anticipation of its translation, no change in white blood cell count was observed in mice and potential benefits may ultimately be found to outweigh the risks. “

3. Table 1, caption, "Characteristics of NHD13 mice are consistent with elevated MCV anemia with expanded hematopoiesis and iron overload which is appropriate as a mouse model of MDS". There exists grammar mistakes in this caption and therefore it is difficult to understand. This should be fixed.

We have modified the caption for Table I to “Hematopoiesis- and Iron-Related Characteristics of NHD13 Mice” in accordance with this comment.

4. Figure 10, the schematic diagram is rather difficult to follow and understand for the readers. The authors should try to highlight the key differences under three conditions. Some details (font size et al.) should be polished as well.

We have modified Figure 10 to simplify and increase clarity in response to the reviewers’ comments, highlighting only the differences in erythroblast iron trafficking between control, MDS, and DFP-treated MDS conditions as the reviewers suggest.

Reviewer #2 (Recommendations for the authors):There are also substantial technical concerns and some aspects that need to be updated to current standards (the manuscript appears to have been written a long time ago):1. The rationale for the choice of deferiprone as the iron chelator is weak, since most of the clinical data were reported with deferasirox. It would be better to do an initial comparison of the two agents to document if they affect anemia and ineffective erythropoiesis differentially (along the lines of Figure 2 and 3E). This would put the DFP data in the paper in a more appropriate context.

Thank you for this request for clarification. While we appreciate the reviewer’s interest in comparing DFP with DFX, this comparison is outside of the scope of our current manuscript and would be more useful in patients after the preclinical study is complete.

The rationale for choosing DFP as the iron chelator for these studies involves several known elements: 1) DFP has a lower iron binding affinity relative to transferrin (Sohn et al., 2008); 2) DFP results in iron transfer from parenchymal cells to increase serum iron, transferrin saturation, and hepcidin responsiveness to iron (Figure 1); and 3) increased hepcidin responsiveness prevents further iron absorption and recycling (Schmidt et al., 2015; Casu et al., 2016). On the foundation of this knowledge, we hypothesize that in addition to decreasing progression of systemic iron overload, DFP could also ameliorate ineffective erythropoiesis in MDS. We anticipate that the specific mechanism is reliant on the ability of DFP to donate iron to transferrin, altering the balance between different transferrin moieties in circulation, resulting in altered iron trafficking within erythroblasts and enhancing erythroblast differentiation.

To clarify our rationale for choosing DFP, we have expanded this part of the Introduction in the revised manuscript.

2. The graph style needs to be updated to change bar graphs to scatterplots with error bars. This also allows display of mouse sex using different symbols (instead of “data not shown”).

We appreciate this comment. Similarly elevated transferrin saturation (a) (n = 3-4 male mice/group and n = 4-6 female mice/group) and hemoglobin (b) (n = 4-6 male mice/group and n = 4-9 female mice/group) are observed in male and female DFP-treated MDS mice. (c) Bone marrow erythroblasts are decreased to a greater degree in male relative to female DFP-treated MDS mice (n = 4-7 male mice/group and n = 8-9 female mice/group). We have added the data on gender-specific measures to new Figure 1 —figure supplement 3, Figure 2 —figure supplement 1, and Figure 3 —figure supplement 1 in the revised manuscript.

We have also remade all figures in accordance with this comment, now presenting scatterplots with error bars.

3. Supplementary figures 3A and 3B use only 2 mice? Which band is FTH?

We appreciate this comment. This experiment was repeated twice with similar results. We present a single gel in Supplementary Figure 3 (Figure 1 —figure supplement 4 in the revised manuscript), now also with a bar noting which band is FTH1, but have reanalyzed the data with both gels together which results in no statistically significant difference. This has been updated in the text.

4. Supplementary figure 4 is very poor quality and cannot be used for quantification.

We appreciate this comment. In response, we opted to measure mRNA expression of the inflammatory marker Saa1 in the liver; new data on Saa1 expression is now included in new Figure 1 —figure supplement 5 of the revised manuscript in addition to STAT3 signaling in the liver, both demonstrating no differences in the inflammatory component of hepcidin regulation.

5. Supplementary figure 9 does not show visible changes in pStat5. With n = 3 mice the changes would have to be large to be significant.

We appreciate this comment and agree with the reviewer that there are no visible changes. We have also attempted to repeat this analysis without succeeding in improving the quality of the pSTAT5 gel. We refocused Supplementary Figure 9 (new Figure 4 —figure supplement 2 in the revised manuscript) on MDS bone marrow samples in response to Reviewer #3, the data providing sufficient confidence that DFP does not significantly alter these signaling pathways.

6. Erfe is the approved gene symbol for erythroferrone https://www.genenames.org/data/gene-symbol-report/#!/hgnc_id/HGNC:26727

We appreciate this information and have changed Fam132b to Erfe in Figures 1 and 9, Supplementary Figure 5 (new Figure 1 —figure supplement 6), and throughout the text in the revised manuscript.

7. Figure 7C: Why was Figure 7C normalized to ProE? Supplementary Figure 13 seems more informative about the lack of differences between WT, MDS and MDS-DFP.

We appreciate this query and have presented the data in Figure 7C (new Figure 7C, 7D, and 7E in the revised manuscript) in a way that is similar to Supplementary Figure 13 (new Figure 7 —figure supplement 1). The evidence is complementary. The Supplementary Figure data provides evidence of Tfr2 expression changes during erythroblast differentiation relative to the stage prior in each model while the Figure 7C-7E data enable an additional appreciation of cumulative differences relative to WT ProE, demonstrating that Tfr2 expression is relatively decreased in BasoE, PolyE, and OrthoE in MDS mouse bone marrow and does not significantly increase in DFP-treated mouse erythroblasts.

8. Figure 10 was not helpful to my understanding of the proposed model in the context of the data in this manuscript. For example, if TfR2 expression does not change between the groups per previous results, what is the evidence that it differentially affects EpoR signaling in this model?

We have modified Figure 10 to simplify and increase clarity in response to the reviewers’ comments, highlighting only the differences in erythroblast iron trafficking between control, MDS, and DFP-treated MDS conditions as the reviewers suggest.

Reviewer #3 (Recommendations for the authors):Below are some suggestions that I believe would strengthen the science and its presentation.1. In certain areas of the Results, the authors appear to overstate their conclusions. For example, on p. 8, the authors state that "… reversal of ineffective erythropoiesis… leads to restored Hamp responsiveness." While the authors show a correlation between improved erythropoiesis and improved Hamp responsiveness to iron stores, the wording here should be modified, as the authors do not prove causation.

We agree with the reviewer and have modified this statement in the revised manuscript to better align the results with the conclusions.

2. Statistical analysis: Paired t tests may not be appropriate for those comparisons that do not involve repeated measures on the same groups. Consider an unpaired T test, or preferably a one-way ANOVA with an appropriate posthoc test. When the same groups are analyzed for multiple parameters (e.g. comparisons of different stages of terminal erythropoiesis in WT, MDS, and DFP-MDS groups), consider an ANOVA test with repeated measures, followed by an appropriate post-hoc test.

We appreciate this comment and have reanalyzed all revised mouse data using one-way ANOVA with multiple comparisons and Tukey post-test analyses when more than 2 groups were compared. This has been edited in the Methods section in the revised manuscript.

3. Figure 1: In addition to showing the non-heme iron content of liver and spleen, it would be valuable to present the histological distribution of iron within these organs.

In accordance with this reviewer’s comment, we provide liver and spleen sections evaluated by immunohistochemistry, stained with Prussian blue to identify intracellular iron stores (Author response image 1). In the liver, this method is insufficiently sensitive to demonstrate differences seen with more quantifiable methods. In the spleen, iron distribution is as expected in the red pulp. While we are not confident that these results are additionally valuable, we could, however, at the reviewer behest, add these figures to the Supplement.

**Author response image 1. sa2fig1:**